# Temporal and spatial assembly of inner ear hair cell ankle link condensate through phase separation

Huang Wang[1,11], Haibo Du[2,3,11], Rui Ren[2], Tingting Du[4], Lin Lin[1], Zhe Feng[5], Dange Zhao[1], Xiaoxi Wei[1], Xiaoyan Zhai[2], Hongyang Wang[6,7], Tingting Dong[4], Jin-Peng Sun[8], Hao Wu [4], Zhigang Xu [2,9] ✉ & Qing Lu [1,4,10] ✉

Stereocilia are actin-based cell protrusions of inner ear hair cells and are indispensable for mechanotransduction. Ankle links connect the ankle region of developing stereocilia, playing an essential role in stereocilia development. WHRN, PDZD7, ADGRV1 and USH2A have been identified to form the so-called ankle link complex (ALC); however, the detailed mechanism underlying the temporal emergence and degeneration of ankle links remains elusive. Here we show that WHRN and PDZD7 orchestrate ADGRV1 and USH2A to assemble the ALC through liquid-liquid phase separation (LLPS). Disruption of the ALC multivalency for LLPS largely abolishes the distribution of WHRN at the ankle region of stereocilia. Interestingly, high concentration of ADGRV1 inhibits LLPS, providing a potential mechanism for ALC disassembly. Moreover, certain deafness mutations of ALC genes weaken the multivalent interactions of ALC and impair LLPS. In conclusion, our study demonstrates that LLPS mediates ALC formation, providing essential clues for understanding the pathogenesis of deafness.

In the inner ear, the hair bundle, which serves as the mechanosensitive organelle, localizes on the top of the hair cell. Hair bundles, composed of actin-based stereocilia, transduce mechanical signals from sound waves into electrical signals through mechanotransduction[1–4]. Mature stereocilia are organized into rows of graded heights, and adjacent stereocilia are interconnected by extracellular links (tip links, top connectors, shaft connectors, and ankle links)[1,2,4,5]. This characteristic staircase-like pattern of stereocilia is critical for hearing transduction[6]. Ankle links, a mesh of thin fibers connecting stereocilia at their basal regions, are indispensable for coordinating stereociliary development and thus are essential for hearing[7]. Temporally, they only exist during the P2 to P12 stages during stereociliary development[7]. However, the

[1]Key Laboratory for the Genetics of Developmental and Neuropsychiatric Disorders, Ministry of Education, Bio-X Institutes, Shanghai Jiao Tong University, Shanghai 200030, China. [2]Shandong Provincial Key Laboratory of Animal Cell and Developmental Biology, School of Life Sciences, Shandong University, Qingdao, Shandong 266237, China. [3]Air Force Medical Center, PLA, Beijing 100074, China. [4]Department of Otolaryngology-Head and Neck Surgery, Shanghai Ninth People's Hospital, Shanghai Jiao Tong University School of Medicine, Shanghai, China. Ear Institute, Shanghai Jiao Tong University School of Medicine, Shanghai, China. Shanghai Key Laboratory of Translational Medicine on Ear and Nose Diseases, Shanghai, China. [5]School of Life Sciences, Fudan University, Shanghai 200433, China. [6]College of Otolaryngology, Head and Neck Surgery, Department of Audiology and Vestibular Medicine, Chinese PLA Institute of Otolaryngology, Chinese PLA General Hospital, Medical School of Chinese PLA, 28 Fuxing Road, 100853 Beijing, China. [7]National Clinical Research Center for Otolaryngologic Diseases, Chinese PLA General Hospital, Medical School of Chinese PLA, 28 Fuxing Road, 100853 Beijing, China. [8]Key Laboratory Experimental Teratology of the Ministry of Education, Department of Biochemistry and Molecular Biology, School of Basic Medical Sciences, Cheeloo College of Medicine, Shandong University, Jinan, China. [9]Shandong Provincial Collaborative Innovation Center of Cell Biology, Shandong Normal University, Jinan, Shandong 250014, China. [10]Bio-X-Renji Hospital Research Center, School of Medicine, Renji Hospital, Shanghai Jiao Tong University, Shanghai, China. [11]These authors contributed equally: Huang Wang, Haibo Du. ✉e-mail: xuzg@sdu.edu.cn; luqing67@sjtu.edu.cn

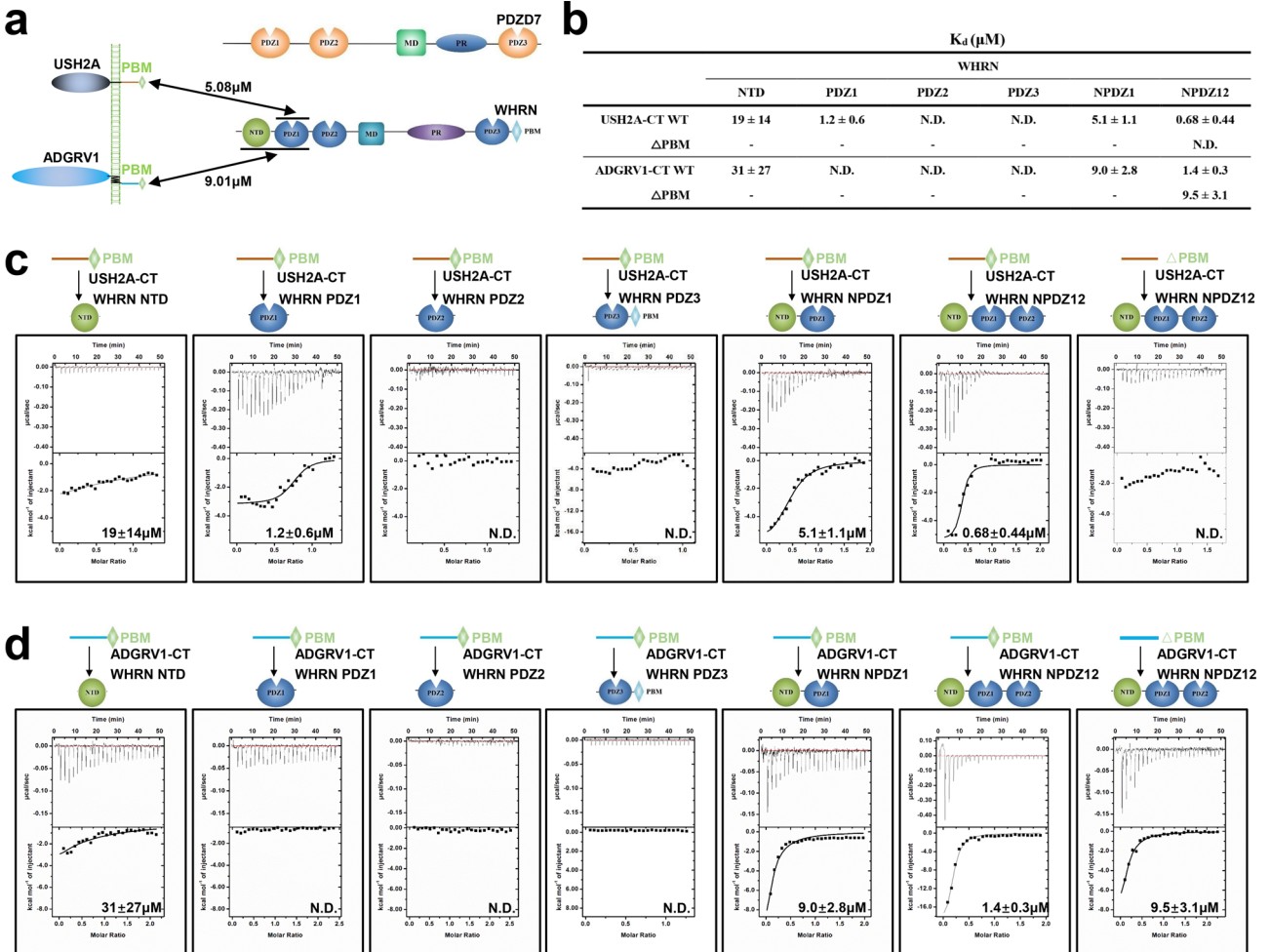

**Fig. 1 | Characterization of the binding selectivity between WHRN with ADGRV1 and USH2A. a** The schematic diagram shows the interactions between WHRN and USH2A-CT, ADGRV1-CT. **b** Summaries of the binding affinities between WT or △PBM constructs of USH2A-CT, ADGRV1-CT, and different WHRN fragments. **c, d** Binding affinities of USH2A-CT (C) or ADGRV1-CT (D) to different WHRN fragments determined by ITC. Truncating PBM (△PBM) from USH2A-CT (C) or ADGRV1-CT (D) abolished or decreased its binding affinity with WHRN NPDZ12. N.D., not detectable. The WHRN fragments include NTD, PDZ1, PDZ2, PDZ3, NPDZ1, and NPDZ12. PBM, PDZ binding motif. NTD, N-terminal domain.

mechanism underlying the emergence and degeneration of ankle links needs to be better understood.

Usher syndrome (USH) is the primary cause of inherited combined vision and hearing deficits[1,2,8–10] and has been clinically classified into three subtypes[10,11]. USH2 is the most prevalent subtype and accounts for 60% of all cases of USH, characterized by moderate to severe congenital hearing loss, some balance problems, and late onset of retinitis pigmentosa by teenage years[12,13]. *WHRN*, *USH2A*, and *ADGRV1* are USH2 causative genes[12,14,15], while *PDZD7* is a USH2 modifier gene[16]. All four USH-associated proteins localize at the basal region of hair cell stereocilia to form the ankle link complex (also known as the USH2 protein complex)[17–20]. Genetic knockout or abnormal expression of any of these four USH2 genes, except for *WHRN*, causes mislocalization of the other three USH2 proteins from the stereociliary ankle-link region[20–25].

Phase separation has been suggested to be essential for USH1 condensate assembly at the tip-link region of stereocilia[26,27]. Similarly, an apparent albeit weaker electron-dense structure can also be observed at the ankle-link region of stereocilia[7,19]. The detailed mechanism of USH2 protein complex assembly in the stereocilia is crucial for understanding the development of the hearing system and the pathogenesis of deafness. WHRN and PDZD7 contain three PDZ domains that recognize specific PDZ binding motifs (PBMs) for

organizing protein networks (Fig. 1a). Meanwhile, both USH2A and ADGRV1 are transmembrane adhesion proteins that form extracellular linkages. At the extreme C-terminus of their cytoplasmic regions, there are type I PBMs, with the consensus amino acid sequence DTHL (Supplementary Fig. 1)[28]. Therefore, PDZ/PBM-mediated multivalent interaction might play essential roles in the USH2 protein complex assembly.

Despite the crucial roles of ankle links in coordinating stereociliary development, several key questions remain to be answered. First, what is the molecular mechanism underlying the assembly and disassembly of the ankle link? Second, how could ankle link proteins contribute to the formation of electron-dense assembly? Finally, what is the relationship between ankle-link condensate formation and the pathogenesis of USH2? In this study, we report that USH2 proteins undergo liquid-liquid phase separation (LLPS) to form ankle link condensates. We find that multivalent interactions in WHRN/USH2A-CT drive the assembly of protein condensates via LLPS, which further recruits PDZD7 and ADGRV1-CT. Interestingly, at high concentrations ADGRV1 inhibits LLPS in vitro, providing a potential mechanism for the temporal disassembly of the ankle link complex. More importantly, utilizing the in vivo mouse model, we show that disruption of the USH2 multivalency for LLPS abolishes the native distribution of the long WHRN at the basal region of stereocilia. Furthermore, point mutations

in USH2 genes found in hearing loss patients weaken the multivalent interactions in the USH2 protein complex and disrupt its phase separation, which may lead to hearing loss.

## Results

### Characterization of the binding selectivity of WHRN with ADGRV1 and USH2A

*WHRN*, *USH2A*, and *ADGRV1* are known USH2 causative genes[12,14,15], so we started with characterizing the interactions between their encoded proteins (Fig. 1a). The full-length WHRN contains an N-terminal domain (NTD), two tandem PDZ domains (PDZ1-2), a middle domain (MD), a proline-rich (PR) region, the third PDZ domain (PDZ3), and a type II PBM. In 2014, Jun Yang and colleagues found that PBMs in the cytoplasmic regions (CT) of USH2A and ADGRV1 interact with WHRN PDZ12[13]. Based on sequence analysis (Fig. S1a), we found that WHRN has an additional N-terminal domain (NTD) preceding PDZ1. The amino acid sequence of the WHRN NTD is highly similar to the Harmonin NTD, which forms a supramodule with PDZ1. The two domains act synergistically in binding to SANS[29]. To investigate the role of the WHRN NTD in PDZ-mediated selective interactions, we generated constructs of various lengths and performed isothermal titration calorimetry (ITC)-based experiments to measure the binding affinities between different WHRN fragments and USH2A-CT or ADGRV1-CT (Fig. 1a). The results revealed that WHRN PDZ1 provides the dominant binding interface for USH2A (Fig. 1a–c). USH2A-CT binds to PDZ1 of WHRN with a binding affinity ($K_d = $ ~1.2 μM) comparable to that of NPDZ1 or NPDZ12 ($K_d = $ ~5.1 or 0.68 μM, respectively). No or very weak binding affinity was detected between USH2A-CT and NTD, PDZ2, or PDZ3 (Fig. 1c). Deletion of PBM from USH2A-CT (△PBM) completely abolished the WHRN NPDZ12/USH2A-CT interaction (Fig. 1c, Supplementary Fig. 1c). Thus, binding between WHRN and USH2A is PDZ/PBM-mediated and does not involve the NTD.

In contrast, ITC-based assays revealed that NTD and PDZ1 are essential for WHRN to interact with ADGRV1 (Fig. 1a, b, and d). ADGRV1-CT exhibited similar binding strengths to NPDZ1 and NPDZ12 and very weak or no interaction with other regions of WHRN (Fig. 1d). However, deletion of ADGRV1-CT PBM only slightly affected its binding with WHRN NPDZ12 ($K_d = $ ~9.5 μM vs. $K_d = $ ~1.4 μM, Fig. 1d, Supplementary Fig. 1d), suggesting that an additional element was involved in the WHRN/ADGRV1 interaction. Harmonin and WHRN are paralog proteins in inner ear hair cells, and both have an NTD domain before PDZ1. Combined with our bioinformatics analysis, we hypothesized that NTD and PDZ1 of WHRN might form a supramodule, NPDZ1, that recognizes ADGRV1, similar to the synergistic binding observed between Harmonin's NTD and PDZ1 to SANS[29]. Therefore, our findings reveal a previously unknown binding mode between WHRN's NTD and ADGRV1. The canonical PDZ/PBM interaction is critical for WHRN binding to USH2A but not for binding to ADGRV1.

### Characterization of the interactions of PDZD7 with ADGRV1 and USH2A

Since *PDZD7* has been reported to be a USH2 modifier gene[16], we next sought to determine how PDZD7 contributes to the assembly of the USH2 complex (Fig. 2a). PDZD7 also contains two tandem PDZ domains (PDZ1-2), an MD, a PR region, and PDZ3. However, compared with WHRN, PDZD7 doesn't have an NTD at the N-terminus. After intensive purification trials, we obtained a shorter USH2A cytoplasmic fragment that is stable as a monomer in solution. We used this fragment USH2A-CT (Short, S) to test its interaction with PDZD7 in ITC-based experiments (Fig. 2a–c). USH2A-CT(S) binds to PDZ1 of PDZD7 with a binding affinity ($K_d = $ ~66 μM) comparable to that of PDZ12 ($K_d = $ ~15 μM). No interaction was detected between USH2A-CT(S) and PDZ2 or PDZ3 (Fig. 2c). Deletion of PBM completely abolished the binding of USH2A-CT(S) to PDZD7 PDZ12, suggesting that PDZ/PBM binding is essential for their interaction (Fig. 2c). Weak association

between ADGRV1 and PDZD7 was reported by using different assay methods[30]. However, ADGRV1-CT displayed no detectable interaction with PDZ12 or PDZ3 of PDZD7 in ITC-based assays (Fig. 2d), and this was further confirmed by GST pull-down assays, which revealed that GST-PDZD7 PDZ12 could pull down USH2A-CT(S) but not ADGRV1-CT (Fig. 2e).

Surprisingly, using ITC to measure the affinity, there was no detectable interaction between WHRN NPDZ12 and PDZD7 PDZ12 or between USH2A-CT and ADGRV1-CT (Fig. 2b, Supplementary Fig. 2a, b), both of which were previously reported as the linkage forming the USH2 protein complex[13]. Since WHRN PDZ1 binds to both USH2A and ADGRV1, there might be competition. GST-WHRN pull-down confirmed that both USH2A-CT(S) and ADGRV1-CT bands were moderately weaker compared to individual groups (i.e., GST-WHRN with USH2A or ADGRV1), suggesting that USH2A and ADGRV1 likely compete for binding to WHRN (Fig. 2f).

### WHRN NTD dimerization contributes to the assembly of the USH2 complex

We next attempted to uncover the potential remaining connections for quaternary USH2 complex assembly. Interestingly, we found that WHRN NTD dimerized and contributed to the assembly of the USH2 complex (Fig. 3a). We conducted size exclusion chromatography coupled with multiangle light scattering (SEC-MALS) experiments to analyze the oligomerization status of purified proteins in solution. Interestingly, Trx-WHRN NPDZ12 remained stable dimer in solution (Fig. 3b, c). Moreover, the NTD alone tended to further oligomerize into larger molecular weight species with increasing protein concentration (Supplementary Fig. 3a).

WHRN dimerization provides the linkage for USH2 complex assembly. When Trx-WHRN NPDZ12 was mixed with Trx-USH2A-CT(S) or Trx-ADGRV1-CT, a complex with a measured mass of ~144.7 or ~149.6 kDa, respectively, was formed on a size-exclusion column. The stoichiometry of both complexes can be fitted as one WHRN dimer binds to two copies of USH2A-CT(S) or ADGRV1-CT (Fig. 3b, c). When the three proteins were mixed, a complex with a detected molecular mass of ~151.6 kDa was eluted, which could be fitted as a 1:2:1 complex composed of one Trx-USH2A-CT(S), two Trx-WHRN NPDZ12 and one Trx-ADGRV1-CT (Fig. 3b, c). The elutions of all three proteins in the complex peak was confirmed by SDS–PAGE (Fig. 3d). Overall, the above findings suggest that USH2A and ADGRV1 simultaneously bind to the dimerized WHRN protein to assemble the ternary USH2 complex.

### Assembly of the quaternary USH2 complex requires USH2A-CT dimerization

As another critical player in the USH2 complex[13,24,31], we found that PDZD7 was recruited by PDZ1-mediated interaction with dimerized USH2A (Fig. 4a). Longer USH2A-CT (a.a. 5086 to a.a. 5193) tended to dimerize (Fig. 4b). In contrast, USH2A-CT(S) (a.a. 5118 to a.a. 5193) existed as a monomer (Supplementary Fig. 3c). USH2A dimerization is necessary for recruiting PDZD7 to quaternary USH2 complex assembly. We conducted two sets of quaternary GST pull-down assays using dimerized USH2A-CT or monomeric USH2A-CT(S). As expected, in the presence of dimerized USH2A-CT, PDZD7 PDZ12 together with USH2A-CT and ADGRV1-CT could be simultaneously pulled down by GST-WHRN NPDZ12 (Fig. 4c). In the control group with monomeric USH2A-CT(S), PDZD7 PDZ12 could no longer be pulled down (Supplementary Fig. 4). Next, we performed a cosedimentation assay and found that all four USH2 proteins were enriched in the pellet fraction (Fig. 4d).

### LLPS of quaternary USH2 protein condensates

All the detailed characterizations of the multivalent interactions among USH2 proteins (PDZs/PBMs, WHRN NTD dimerization, and USH2A dimerization) and the cosedimentation of four proteins in the

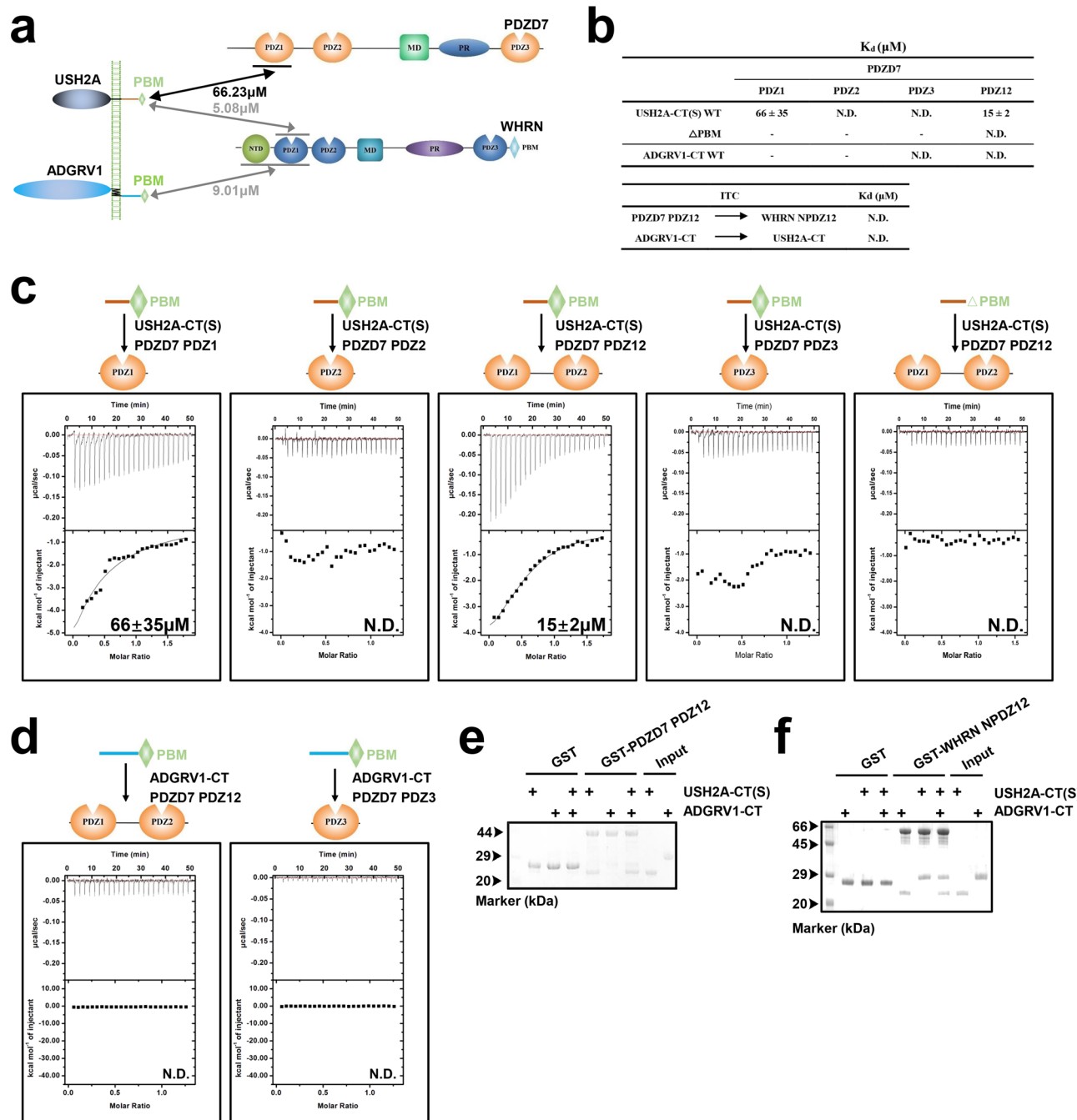

**Fig. 2 | Characterization of the interactions between PDZD7 with ADGRV1 and USH2A. a** The schematic diagram shows the interactions among the four USH2 proteins. **b** Up panel, summaries of the binding affinities between WT or △PBM constructs of USH2A-CT(S) or ADGRV1-CT and different PDZD7 fragments; bottom panel, summaries of the binding affinities between PDZD7 PDZ12 and WHRN NPDZ12, ADGRV1-CT and USH2A-CT. **c** Binding affinities of USH2A-CT(S) to different PDZD7 fragments determined by ITC. Truncating PBM (△PBM) from USH2A-CT abolished the binding to PDZD7 PDZ12. **d** Binding affinities of ADGRV1-CT to different PDZD7 fragments determined by ITC. **e** GST pull-down assay showing that GST-PDZD7 PDZ12 pulled down Usherin-CT(S) but not VLGR1-CT. **f** GST pull-down assays show that GST-WHRN NPDZ12 pulled down USH2A-CT(S) and ADGRV1-CT. N.D., not detectable. The different PDZD7 fragments include PDZ1, PDZ2, PDZ2, PDZ12, or PDZ3. PBM, PDZ binding motif. Source data are provided as a Source Data file.

pellet strongly suggest the occurrence of LLPS. To dissect this process, we mixed WHRN NPDZ12 with USH2A-CT. More proteins were recovered from the condensed phase (pellet) compared to the isolated group (Fig. 5a). When WHRN NPDZ12 and USH2A-CT proteins were mixed, the formation of spherical and liquid-like droplets with enrichments of both proteins was observed under the fluorescence microscope, but not on their own (Fig. 5b). The LLPS capacity of WHRN NPDZ12/USH2A-CT was positively correlated with their protein concentration, whereby increased protein condensation was observed

with increasing protein concentrations (Supplementary Fig. 5a, b). WHRN NPDZ12/ADGRV1-CT exhibited little protein condensation after mixing, suggesting that the formation of the LLPS platform is only mediated by WHRN/USH2A (Supplementary Fig. 6a).

Multivalency is acknowledged as a driving force for LLPS; therefore, we investigated the impact of USH2 multivalent interactions on phase separation using USH2A-CT △PBM to disrupt the PDZ/PBM interaction, WHRN NPDZ12 △NTD to eliminate NTD dimerization, and USH2A-CT(S) to weaken USH2A dimerization. All these truncations

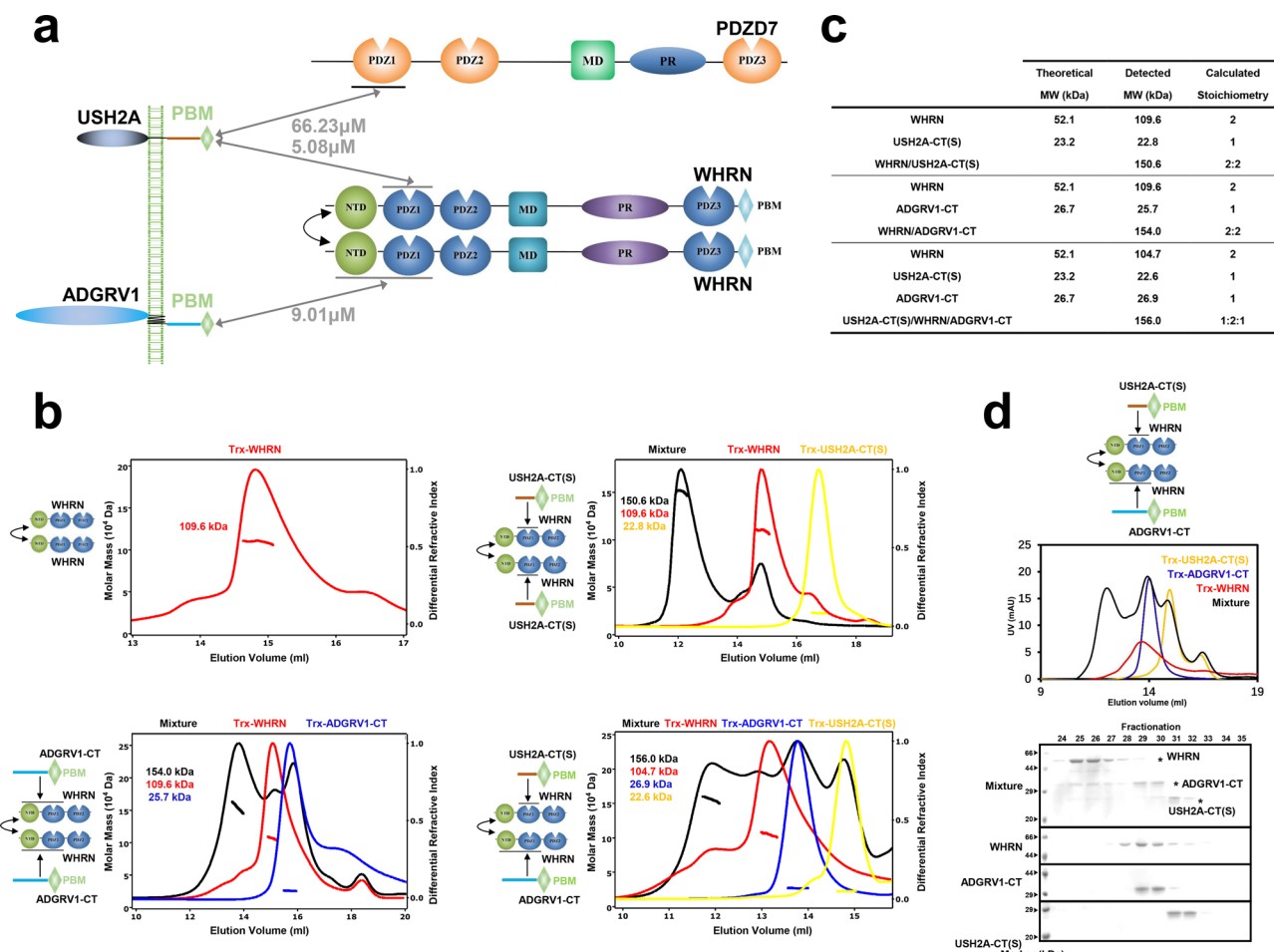

**Fig. 3 | WHRN NTD dimerization contributes to the assembly of the ternary USH2 complex. a** The schematic diagram shows that dimerized WHRN interacts with USH2A and ADGRV1 to assemble a ternary USH2 complex. **b** SEC-MALS showed the molecular sizes of WHRN, WHRN/USH2A-CT(S), WHRN/ADGRV1-CT, or USH2A-CT(S)/WHRN/ADGRV1-CT. The samples for fitting molecular sizes were eluted on a Superose 12 10/300 GL or Superdex 200 increase 10/300 GL column. **c** Summaries of stoichiometry in **b**. **d** Fractionation coupled with FPLC showed three bands on the gel, confirming the existence of WHRN, USH2A-CT(S), and ADGRV1-CT in the peak. WHRN NPDZ12 was abbreviated as WHRN. Source data are provided as a Source Data file.

were expected to decrease the level of multivalency in the WHRN/USH2A-CT complex, which might interfere with USH2 condensation (Fig. 5c). As expected, these mutants all demonstrated much weaker phase separation capacity (i.e., lower levels of protein enrichment in the pellet fraction and reduced droplet formation under a fluorescence microscope) compared to the WT protein, suggesting that multivalent interactions (PDZs/PBMs, WHRN NTD dimerization, and USH2A dimerization) are crucial for USH2 LLPS (Fig. 5d, e, and f).

WHRN NPDZ12/USH2A-CT-enriched condensates further recruited PDZD7 PDZ12 and ADGRV1-CT (Fig. 5g). The small droplets gradually fused into larger droplets (Supplementary Fig. 6b). Fluorescence recovery after photobleaching (FRAP) analysis of the proteins in the droplets showed that the signals of all the four components recovered after photobleaching, indicating that protein constituents can freely exchange between the condensate droplets and the surrounding diluted solution (Fig. 5h and Supplementary Fig. 6c). The dynamic nature of the droplets confirms that protein fractions inside are not immobile aggregates. Thus, we reveal that multivalent interactions among USH2 proteins drive the formation of quaternary USH2 protein condensates mediated by LLPS.

## A high concentration of ADGRV1-CT disassembles the USH2 condensates formed by phase separation

Since the WHRN NTD is required both for binding to ADGRV1-CT and phase separation with USH2A-CT, there might be potential competition

between these two events. Therefore, we hypothesized that competition with NTD may disturb the multivalent interaction for LLPS and disrupt USH2 condensate formation. To test this competing effect, we added 0, 2, 10, 20, or 60 µM ADGRV1-CT to the WHRN NPDZ12/USH2A-CT protein mixture (20 µM each). We found that the pellet enrichment of WHRN NPDZ12 and USH2A-CT decreased with increasing ADGRV1-CT concentration (Fig. 6a and Supplementary Fig. 7a). Consistently, under a fluorescence microscope, the droplets formed by WHRN NPDZ12/USH2A-CT decreased in amount and size in an ADGRV1-CT concentration-dependent manner (Fig. 6b). To exclude the involvement of PDZ/PBM binding in the ADGRV1-mediated disassembly of condensates, we deleted PBM from ADGRV1-CT. The cosedimentation assay revealed that the pellet enrichments of WHRN NPDZ12 and USH2A-CT remained decreased (Fig. 6c). Additionally, deletion of PBM from ADGRV1-CT did not change the amounts or sizes of the droplets formed by WHRN NPDZ12/USH2A-CT (Fig. 6d). This suggests that the disassembly of condensates by ADGRV1 is not achieved through PDZ/PBM binding with WHRN.

To further validate our hypothesis, we built the complex structure model of ADGRV1-CT with WHRN NTD using the Alphafold prediction tool, with the pLDDT value calculated to be 87.86 (https://studio.hpc.sjtu.edu.cn). Consistent with our prediction, the WHRN NTD adopts a compact five-helix-bundle structure similar to the Harmonin NTD (Fig. 6e, left panel). In the NTD helix bundle, αA and αB form a V-shaped helix hairpin. Importantly, in this hairpin region, the side

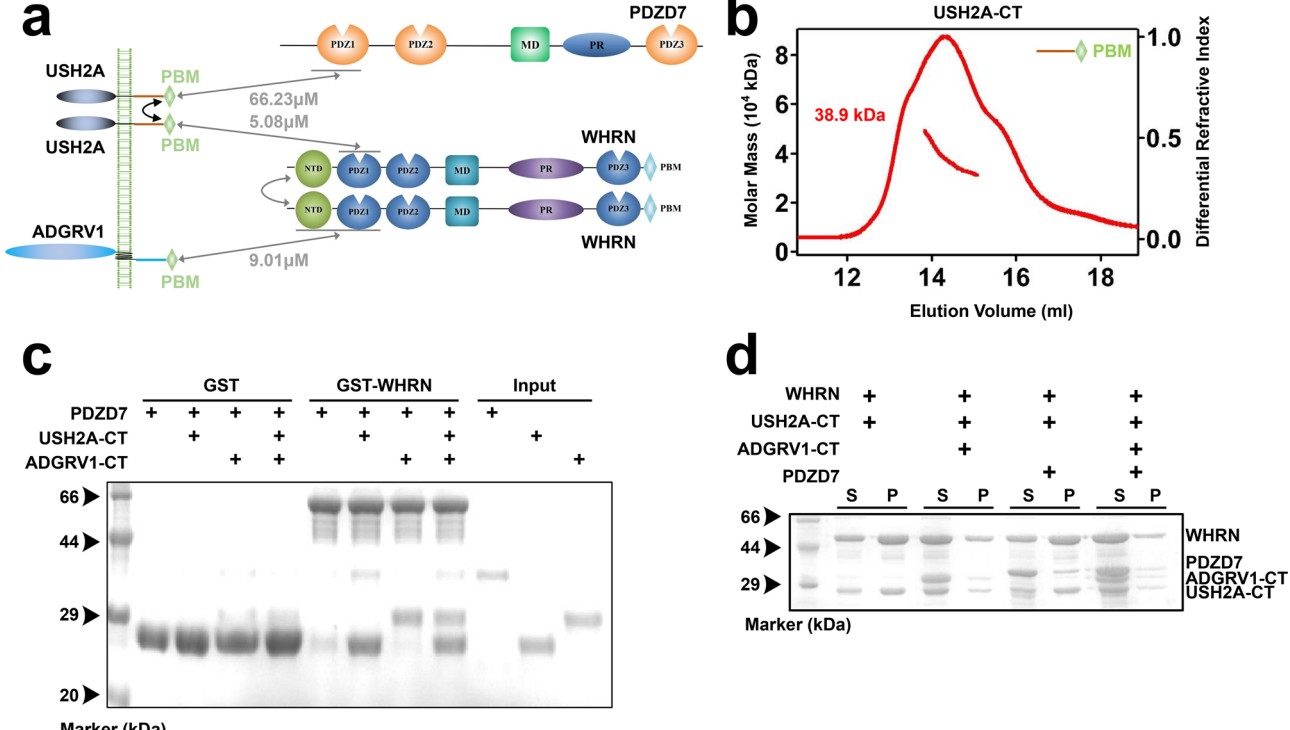

**Fig. 4 | The assembly of the quaternary USH2 complex requires the dimerized USH2A-CT. a** The schematic diagram shows that PDZD7 is recruited by dimerized USH2A to assemble the quaternary USH2 protein complex. **b** SEC-MALS showing the molecular size of USH2A-CT, summarized in Fig. 5c. **c** GST pull-down assay showing that GST-WHRN pulled down PDZD7 in the presence of USH2A-CT and quaternary protein complex was successfully assembled. **d** Co-sedimentation assay showed that WHRN, PDZD7, ADGRV1-CT, and USH2A-CT were enriched in the pellet. WHRN NPDZ12 and PDZD7 PDZ12 were abbreviated as WHRN, and PDZD7, respectively. Source data are provided as a Source Data file.

chains of V39, L42, L46, L50, F58, and L62, which are conserved among species (Supplementary Fig. 1a), form a hydrophobic cleft (Fig. 6e, middle panel, indicated by the dashed line). In the ADGRV1-CT structure, a short stretch of ~20 amino acids (hereafter referred to as the NTD binding motif (NBM)) upstream of the PBM adopts an α-helical conformation. It occupies the cleft of the αA/αB helix-hairpin in the WHRN NTD to establish hydrophobic contacts (Fig. 6e, right panel). The superimposition of the ADGRV1-CT/WHRN NTD complex structure with the Cad23-CT/Harmonin NTD complex structure shows a similar binding mode whereby the hydrophobic cleft is occupied by NBM (Supplementary Fig. 7b). More importantly, deletion of NBM from ADGRV1-CT abolished its binding to the WHRN NTD, highlighting the reliability of our predicted complex structure (Supplementary Fig. 7c). Compared to the WT protein, ADGRV1-CT △NBM exhibited decreased binding affinity for WHRN NPDZ12, as PBM contributed partly to the interaction (Supplementary Fig. 7c). This previously unrevealed interaction model reinforces the critical role of NTD in USH2 condensate formation. The high concentration of ADGRV1-CT inhibits USH2 condensate formation by disrupting NTD oligomerization, providing a possible explanation for ankle link disassembly.

### NTD is necessary for WHRN to target the ankle region of stereocilia

To further confirm the essential role of NTD of WHRN in ALC formation, we injectoporated expression plasmids encoding full-length WHRN or WHRN without NTD domain into hair cells and examined their localization using a confocal microscope. The results show that EGFP-tagged full-length WHRN clearly targets the tips as well as the ankle region of stereocilia (Fig. 7a), in a pattern similar to endogenous WHRN[24,25]. At the ankle region of stereocilia, EGFP-WHRN is colocalized with ADGRV1 as expected (Fig. 7a). However, the stereociliary ankle localization is significantly reduced in EGFP-tagged WHRN without NTD domain (Fig. 7b), suggesting that this domain is essential for WHRN to target the ankle region of stereocilia. Meanwhile, EGFP-tagged WHRN without NTD domain could still target the stereociliary tips efficiently (Fig. 7b). Taken together, our data shows that NTD is necessary for WHRN to target the ankle region of stereocilia but not the stereociliary tips.

### Deafness-related mutations affect the LLPS capacities of USH2 proteins

USH2 genes are associated with USH2 syndrome and nonsyndromic hearing loss. According to the Human Gene Mutation Database (www.hgmd.cf.ac.uk/) and the ClinVar database (https://www.ncbi.nlm.nih.gov/clinvar/), a total of 7 missense and frameshift mutations involving USH2 condensate formation have been documented. The distributions of these missense variants were mapped onto the structure models of WHRN NPDZ1 and PDZD7 PDZ1 (Fig. 8a). Mutations at the corresponding positions were also tested for their impacts on phase separation of the USH2 condensates. Among the 7 mutation sites, 3 are on WHRN NPDZ1, 1 on ADGRV1-CT, and 3 on PDZD7 PDZ12.

With respect to WHRN NPDZ1, there are three deafness-related disease mutations: A64D, T110A, and R223H (NCBI ClinVar database, VCV000045664, VCV000178844, and VCV000045681). According to sequence alignment, A64, T110, and R223 are evolutionarily invariant (Supplementary Fig. 1a). USH2A-CT(S) binds to WHRN NPDZ12 T110A with a weaker binding affinity ($K_d$ = ~16 μM) than that of the WT ($K_d$ = ~2.7 μM), which may contribute to deafness (Fig. 8b and Supplementary Fig. 8a). A64D and R223H exert no effect on the WHRN NPDZ12/USH2A-CT(S) interaction, given that their $K_d$ values were comparable to that of the WT (5.5 μM, 4.7 μM vs. 2.7 μM, Fig. 8b and Supplementary Fig. 8a). However, both mutations weakened the condensation of USH2 proteins (Fig. 8d, e). The molecular size of WHRN NPDZ12 A64D was

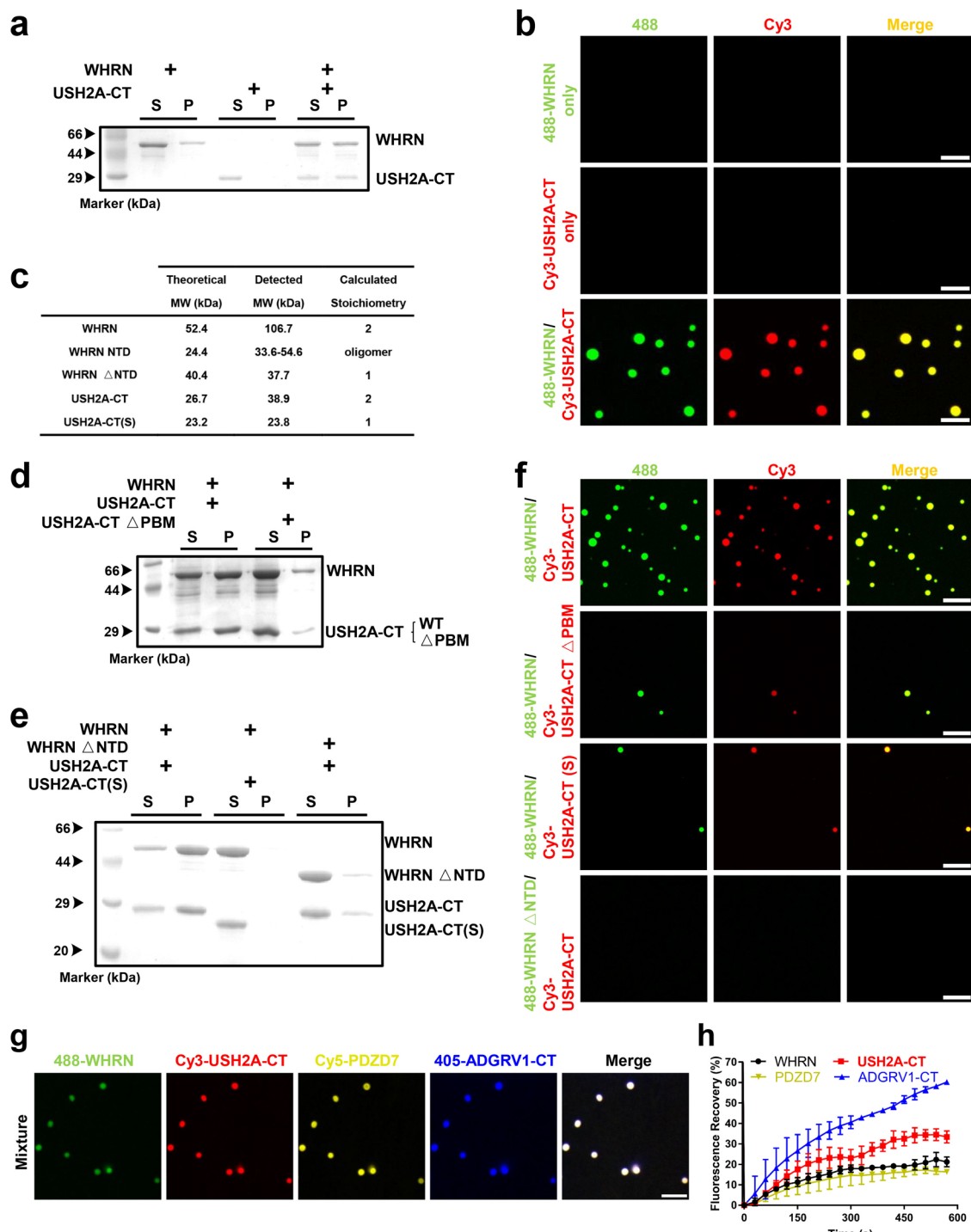

**Fig. 5 | Liquid-Liquid Phase Separation (LLPS) of quaternary USH2 protein condensate. a** Co-sedimentation-based assay showing that proteins were much more enriched in the pellet in WHRN/USH2A-CT than the isolated group. **b** Fluorescence images showing that droplets formed in WHRN/USH2A-CT but not the isolated group (20 μM each). WHRN and USH2A-CT were labeled with Alexa 488 and Cy3, respectively. Labeled proteins were added at a ratio of 1%. Scale bar: 5 μm. **c** Summaries of the stoichiometry of WHRN, WHRN NTD, WHRN △NTD, USH2A-CT, or USH2A-CT(S). **d** Co-sedimentation-based assay showing that the pellet enrichments of the two proteins in WHRN/USH2A-CT were deceased when USH2A-CT △PBM was introduced. Proteins were mixed at the final concentration of 20 μM each. **e** Co-sedimentation-based assays showed that the pellet enrichments of the two proteins in WHRN/USH2A-CT were decreased when WHRN △NTD or USH2A-CT(S) was introduced. Proteins were mixed at the final concentration of 20 μM

each. **f** Fluorescence images show that the droplets formed by WHRN/USH2A-CT decreased in size or number when USH2A-CT △PBM, USH2A-CT(S), or WHRN △NTD was introduced. Proteins were mixed at the final concentration of 20 μM each. Scale bar: 10 μm. **g** Fluorescence images showed that the droplets were colocalized in highly enriched WHRN, USH2A-CT, PDZD7, and ADGRV1-CT. WHRN, USH2A-CT, PDZD7, and ADGRV1-CT were labeled with Alexa 488, Cy3, Cy5, and Alexa-405, respectively. Labeled proteins were added at a ratio of 1%. Scale bar: 5 μm. **h** Kinetic quantification of the FRAP assay showing the recovery rates of WHRN, USH2A-CT, ADGRV1-CT, and PDZD7. Mean ± SD, *n* = 3–5. WHRN NPDZ12 and PDZD7 were abbreviated as WHRN and PDZD7, respectively. △PBM or △NTD, the truncation of PBM or NTD. PBM, PDZ binding motif. NTD, N-terminal domain. Source data are provided as a Source Data file.

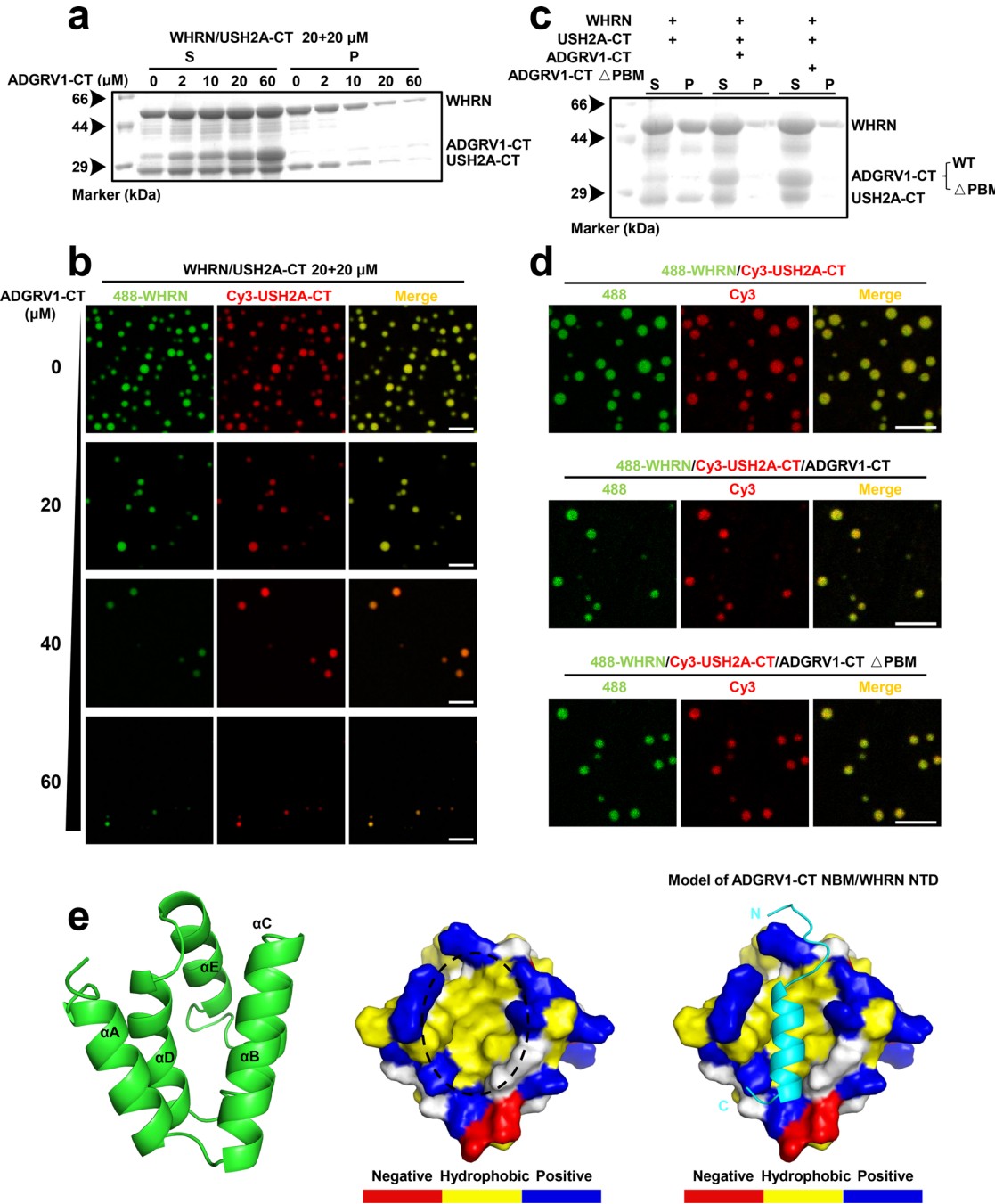

**Fig. 6 | High concentration of ADGRV1-CT disassembles the USH2 condensates formed by phase separation. a** Co-sedimentation-based assay showing that the pellet enrichments of both WHRN and USH2A-CT in WHRN/USH2A-CT (20 μM each) were decreased by ADGRV1-CT concentration-dependently at 0–60 μM. **b** Fluorescence images showing the droplets formed in WHRN/USH2A-CT decreased in both number and size by ADGRV1-CT concentration-dependently at 0–60 μM. WHRN and USH2A-CT were labeled with Alexa 488 and Cy3, respectively. Scale bar: 5 μm. **c** Co-sedimentation-based assay showed that the decrease in the pellet enrichment of WHRN, USH2A-CT, or ADGRV1-CT was not much changed when PBM was truncated (△PBM) from ADGRV1-CT. **d** Fluorescence images show that the amounts and sizes of the droplets formed by WHRN/USH2A-CT was still

decreased by ADGRV1-CT when its PBM was truncated (△PBM). Scale bar: 5 μm. **e** Left panel, stereoview of the predicted structure of WHRN NTD drawn in the ribbon diagram; middle panel, surface representation showing the existence of an exposed hydrophobic pocket in NTD (highlighted by a black circle), which is predicted to serve as the binding site for the NBM of ADGRV1-CT.; right panel, surface representation of the predicted three-dimensional structure of WHRN NTD in complex with the ADGRV1-CT NBM. The hydrophobic, positively charged, negatively charged, and uncharged polar amino acid residues are shown in yellow, blue, red, and gray, respectively; ADGRV1-CT NBM is drawn in the ribbon model in cyan. WHRN NPDZ12 was abbreviated as WHRN. PBM, PDZ binding motif. NBM, NTD binding motif. Source data are provided as a Source Data file.

calculated as 60.9 kDa, suggesting that A64D-mediated dispersion of USH2 condensates was caused by disrupting WHRN dimerization (Fig. 8c). However, of note, WHRN NPDZ12 R223H even increased in molecular size (117.8 vs. 106.7 kDa, data not shown). None of the three disease mutations affected the WHRN NPDZ12/ADGRV1-CT interaction,

with comparable $K_d$ values between mutants and WT (0.41 μM, 0.55 μM, 3.3 vs. 1.4 μM, Fig. 8b and Supplementary Fig. 8b).

Y6236X on ADGRV1-CT introduces a premature stop codon, which leads to the removal of the C-terminal PBM and its upstream sequence, both of which are required for PDZ binding (Supplementary

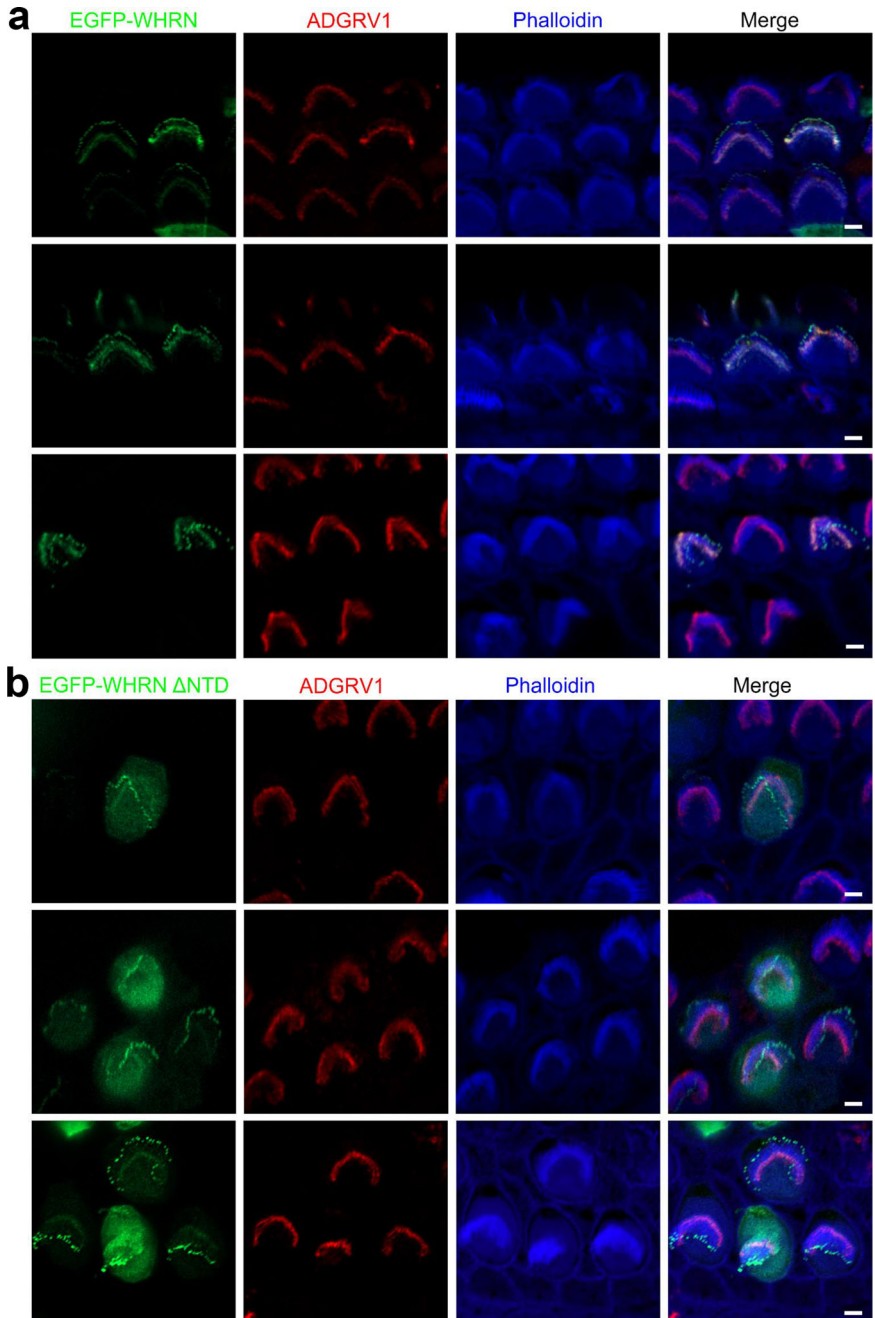

**Fig. 7 | NTD is necessary for WHRN to target the ankle region of stereocilia.** Cochlear explants from P4 wild-type mice were injectoporated with expression vectors to express EGFP-tagged full-length WHRN (**a**) or WHRN lacking NTD (**b**) in cochlear hair cells. Immunostaining with an anti-ADGRV1 antibody was performed to locate ADGRV1 (red). Phalloidin staining was performed to indicate the stereocilia (blue). Scale bars: 2 µm. NTD, N-terminal domain.

Fig. 1d)[14]. As expected, the Y6236X mutant failed to bind WHRN NPDZ12, as demonstrated by ITC-based assays and GST pull-down experiments (Fig. 8b and Supplementary Fig. 8b) (Fig. 9a). Consequently, ADGRV1-CT Y6236X was no longer recruited by the USH2 protein condensates (Fig. 9b, c). On the other hand, inhibition of LLPS from high concentration ADGRV1-CT was disrupted by Y6236X, as the pellet enrichments of WHRN NPDZ12, USH2A-CT, and PDZD7 PDZ12 were no longer decreased (Fig. 9b). Thus, Y6236X may cause deafness by disrupting both assembly of the quaternary USH2 protein complex and inhibition of phase separation.

On PDZD7 PDZ12, there are also three deafness-related disease mutations: G103R, G228R, and M285R (NCBI ClinVar database, VCV000545399, VCV000545403, and VCV000545400). According to

sequence alignment, all three loci are invariant throughout evolution (Supplementary Fig. 1b). ITC-based assays and GST pull-down assays revealed that only the G103R mutation abolished the interaction between PDZD7 PDZ12 and USH2A-CT(S). In contrast, neither the G228R nor M285R mutation had an effect (Fig. 10b and Supplementary Fig. 8c) (Fig. 10c). Consequently, the PDZD7 PDZ12 G103R mutant was no longer enriched in the condensed droplets of USH2 proteins (Fig. 10d). We conclude that G103R disrupts quaternary USH2 protein complex assembly and phase separation by eliminating the interaction between PDZD7 and USH2A, which may be one of the mechanisms underlying deafness.

Here, we combined biochemical and structural approaches to provide mechanistic explanations for these documented

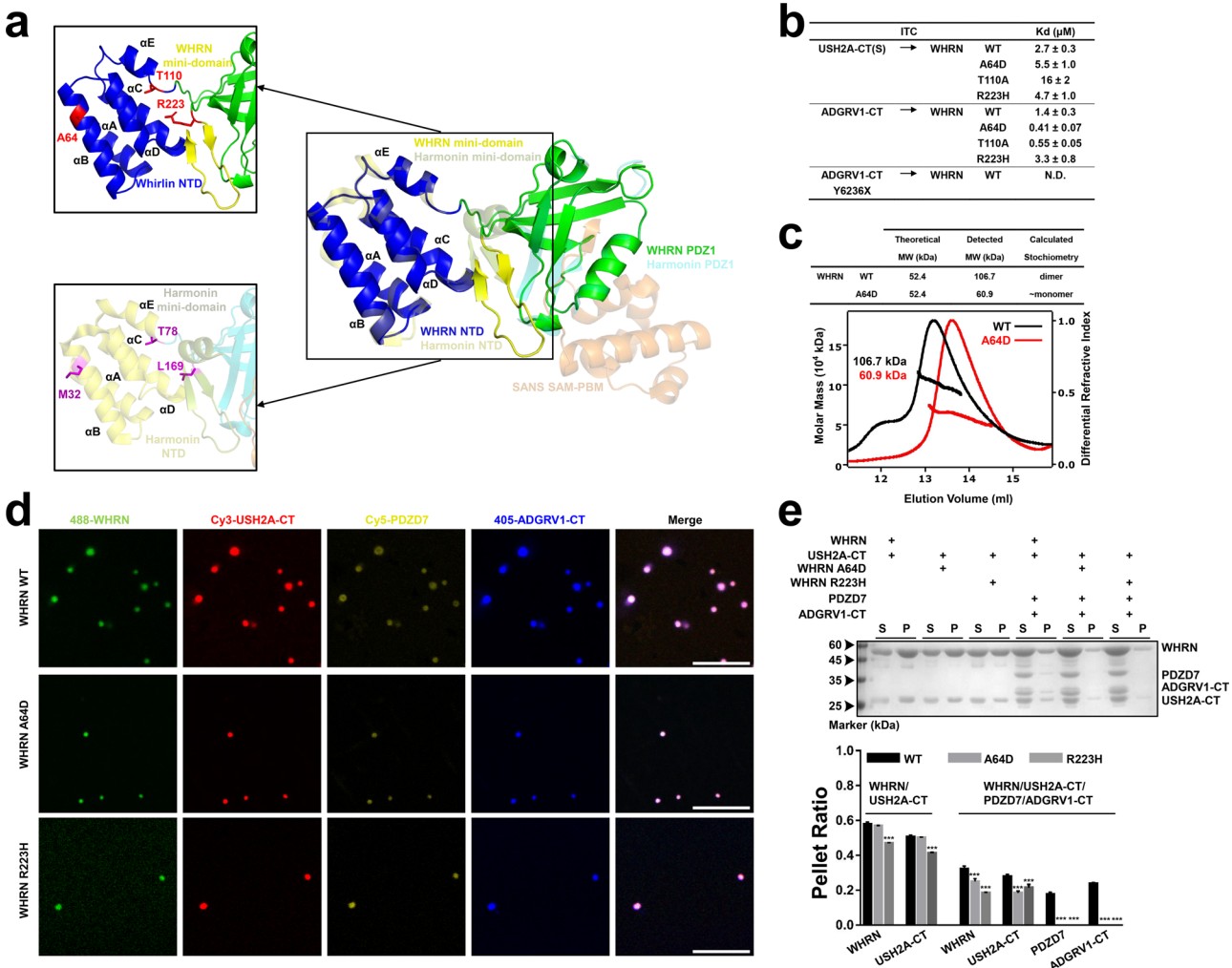

**Fig. 8 | Deafness-related mutations on WHRN affect the LLPS capacities of USH2 proteins. a** The predicted WHRN NPDZ1 structure is superimposed with the resolved structure of Harmonin NPDZ1/SANS SAM-PBM[29]. Superimposition shows that A64, T110, and R223 on WHRN NPDZ1 correspond to M32, T78, and L169 on Harmonin NPDZ1, respectively. The side chain of WHRN A64 points outwards. A64D may disturb WHRN dimerization by disrupting WHRN NTD oligomerization. WHRN T110 localizes to the αE helix of NTD. We speculate that the T110A may affect WHRN's interaction with USH2A by disrupting the overall conformation of WHRN. WHRN R223 localizes on the mini-domain. We speculate that R223H disrupts the overall folding of the mini-domain and further leads to the misfolding of NPDZ1 on WHRN. **b** Summaries of the binding affinities between WT or deafness-related mutants of WHRN, ADGRV1-CT, and USH2A-CT. N.D. not detectable. **c** Up panel, a summary of the SEC-MALS result in the bottom panel; bottom panel, SEC-MALS assay showing that A64D disrupted the dimer capacity of WHRN. **d** Fluorescence images show that A64D and R223H decreased the size and number of the droplets formed by quaternary USH2 protein phase separation. Scale bar: 10 μm. **e** Up panel: Co-sedimentation assay showing that both A64D and R223H decreased the pellet enrichments of WHRN/USH2A-CT/PDZD7/ADGRV1-CT. Bottom panel: Quantifications of each protein recovered from the condensed phase (pellet) in the co-sedimentation assays described in the up panel. Mean ± SD, $n = 3$. ***$p < 0.001$ using one-way ANOVA with Dunnett's multiple comparison test. WHRN NPDZ12 and PDZD7 PDZ12 were abbreviated as WHRN and PDZD7, respectively. Source data are provided as a Source Data file.

disease-causing mutations. Mutations that can cause the impaired assembly of the USH2 complex also weaken the condensed phase formation of the complex. This process may lead to impairments in hearing and vision due to defects in stereociliary development.

## Discussion

Development of the characteristic architecture of stereocilia relies on the coordination of ankle links, which are anchored to stereocilia by the USH2 proteins WHRN, PDZD7, ADGRV1, and USH2A. The underlying mechanisms for the temporal emergence and degeneration of ankle links remain elusive. By performing detailed biochemical and structural experiments, we elucidated the molecular mechanism underlying the assembly of the USH2 protein complex. We uncovered and further characterized the multivalent interactions among USH2 proteins and elucidated LLPS-mediated USH2 condensate formation. Interestingly, at high concentrations ADGRV1 inhibits phase separation

in vitro. We used an in vivo mouse model to dissect the mechanisms underlying the precise distribution of the full-length WHRN to the basal region of stereocilia. Moreover, we found some deafness-related mutations in USH2 genes may cause deafness by disrupting phase separation and condensate assembly.

HHD domains are present in large proteins and manifest various features[32]. The structures and functions of some HHD domains in hearing-related proteins have been reported. Harmonin HHD1 was reported to form a supramodule with its PDZ1, with the two domains acting synergistically in binding to SANS[29]. PDZD7 HHD (denoted as MD in the present study) mediates membrane targeting[33]. There are two HHDs on WHRN: HHD1 and HHD2. The structure of the WHRN HHD2 has been reported (denoted as MD in the present study)[34]. However, little is known about HHD1 (denoted as NTD in the present study). Therefore, the WHRN NTD explored here extends our knowledge of HHD functions. The PDZD7/USH2A interaction is weak, but the

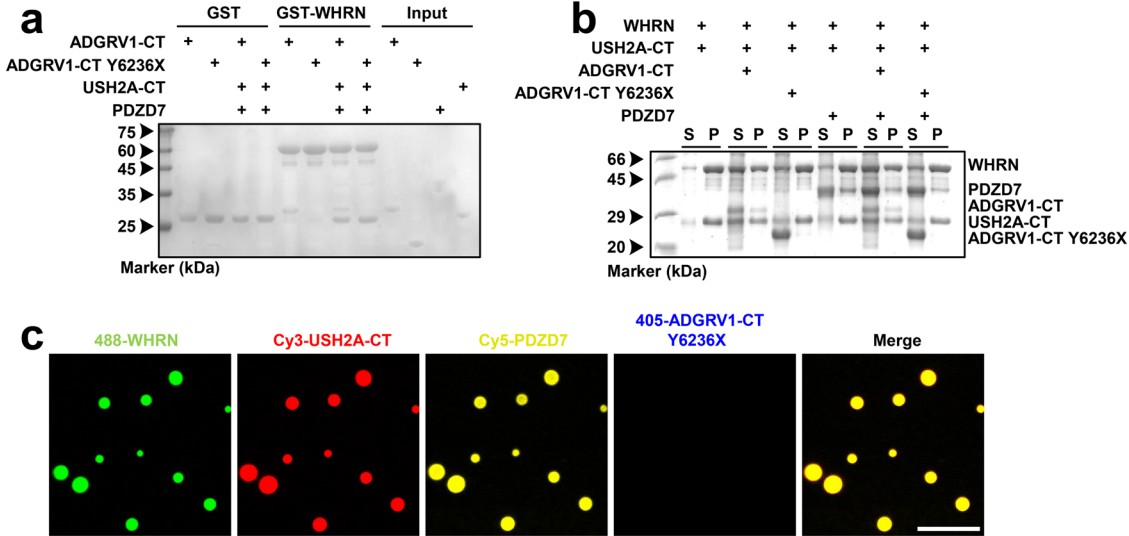

**Fig. 9 | Deafness-related mutation on ADGRV1-CT affects the LLPS capacities of USH2 proteins. a** GST pull-down assay showing that though the pulled-down USH2A-CT and PDZD7 remained unaffected, ADGRV1-CT Y6236X was not pulled down by GST-WHRN and the assembly of quaternary USH2 protein complex was disrupted. **b** Co-sedimentation assay showing that ADGRV1-CT Y6236X was not enriched in the protein pellets, and the protein pellet enrichments of WHRN, USH2A-CT, and PDZD7 increased in the presence of Y6236X when compared to the WT. **c** Fluorescence images show that ADGRV1-CT Y6236X was absent in the droplets formed by USH2 protein phase separation. Scale bar: 10 μm. WHRN NPDZ12 and PDZD7 PDZ12 were abbreviated as WHRN and PDZD7, respectively. Source data are provided as a Source Data file.

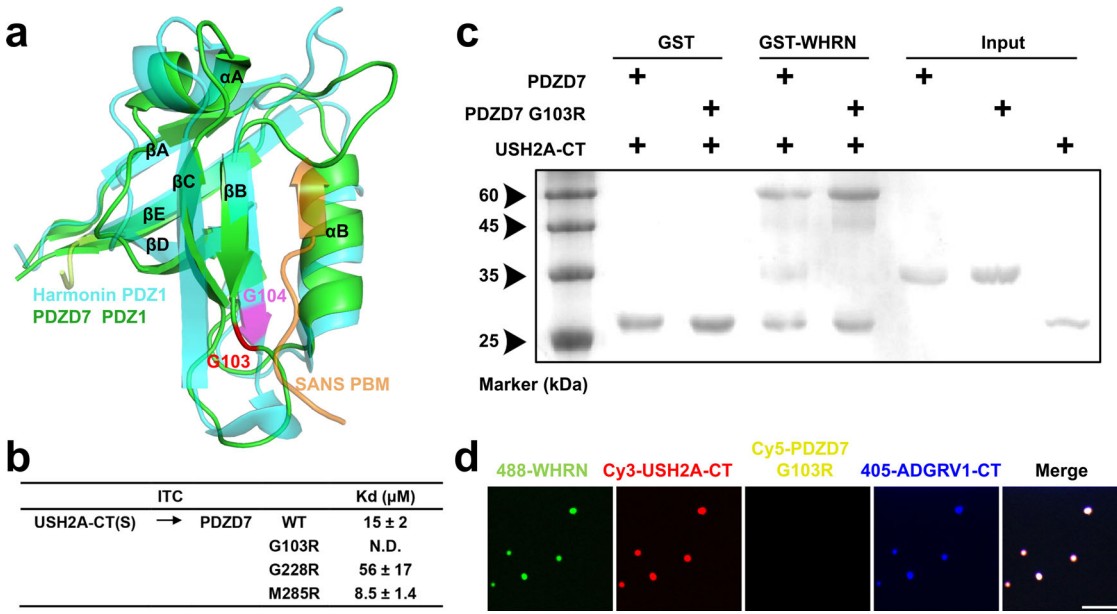

**Fig. 10 | Deafness-related mutation on PDZD7 affects the LLPS capacities of USH2 proteins. a** PDZD7 PDZ1 is structurally superimposed with Harmonin PDZ1/ SANS PBM[29]. Superimposition shows that G103 on PDZD7 PDZ1 corresponds to G104 on Harmonin PDZ1. G103 localizes on the binding groove and may be involved in the interaction with USH2A. **b** Summaries of the binding affinities between WT or deafness-related mutants of PDZD7 and USH2A-CT(S). N.D., not detectable. **c** GST pull-down assay showing that GST-WHRN could not pull down PDZD7 G103R in the presence of USH2A-CT. **d** Fluorescence images show that PDZD7 G103R was absent in the droplets formed by USH2 protein phase separation. Scale bar: 10 μm. WHRN NPDZ12 and PDZD7 PDZ12 were abbreviated as WHRN and PDZD7, respectively. PBM, PDZ binding motif. Source data are provided as a Source Data file.

lipid binding capacity of PDZD7 MD can enrich PDZD7 at the inner surface of the cell membrane[33]. When it binds to the cell membrane, the three-dimensional system will be changed to two dimensions, in which the weak binding is sufficient for micro-cluster formation.

Electron densities have been observed at both the tip-link[26,27] and ankle-link regions of stereocilia[13,22]. Some previous studies have reported that tip-link densities might be formed by phase separation[26,27]. We demonstrate the ankle-link densities to be formed via the phase separation of USH2 proteins, which is essential for stabilizing ankle links and coordinating stereociliary development. The ankle link differs from other types of links connecting stereocilia in that it exists during the P2 to P12 stage, but little is known about its emergence or degeneration. Interestingly, we found that ADGRV1 at high concentrations inhibited the phase separation of the USH2 protein complex. This may explain the degeneration of the ankle links after P12. Notably, full-length WHRN can undergo phase separation by itself[27]. However, due to poor stability and homogeneity of purified protein, our present study adopts the NPDZ12 fragment of WHRN but

not full-length. Therefore, the threshold concentration for phase separation would be lower if full-length WHRN is taken up for the USH2 protein complex.

Localization of WHRN at the tip of stereocilia is determined by its interaction partners, including EPS8 and Myo15[35,36]. This process also works for precise targeting of WHRN at the ankle link region of stereocilia[22,24]. The longer WHRN is targeted to both the ankle-link and tip-link regions of stereocilia, whereas the shorter isoform lacking NTD, PDZ1, and PDZ2 is localized at the tip-link region[25]. This mechanism comes to light when we refer to our in vivo results based on the mouse model. We found NTD is responsible for recruiting WHRN to the ankle-link region, likely through interactions with USH2A and ADGRV1. More importantly, our in vivo mouse model confirms their importance by demonstrating that the mutant of longer WHRN lacking NTD lose their localization at the ankle-link but not the tip-link region. Given our results, the shorter WHRN localizes at the stereociliary tip-link but not the ankle-link region, as it lacks NTD[25].

Biochemical characterization of LLPS and structural analysis provide alternative angles for investigating the molecular mechanisms underlying the pathogenesis of deafness-related mutations. There are two categories. The first category is that the mutations considerably affect the structure conformation or localize on the binding interface, disrupting the target association and damaging the multivalency of LLPS. G103R on PDZD7 abolishes its interaction with USH2A and therefore disrupts the condensation of the USH2 protein complex. Consistently, through superimposition, PDZD7 G103 is mapped to Harmonin G104; both of them localize at the binding groove, making it possible for them to mediate interactions with the binding partner (Fig. 10a). ADGRV1-CT Y6236X, which lacks of the PBM, lost the association with WHRN PDZ1. The effects on phase separation of these mutants (T110A on WHRN, Y6236X on ADGRV1-CT, and G103R on PDZD7) are highly correlated with the impacts of the mutations on their quantitative binding affinity reductions in binding between USH2 proteins, indicating that the LLPS capacities of USH2 complexes are very sensitive to the binding affinities of the respective multivalent interaction network of the assemblies.

Some other mutations, with minor perturbations on protein structures, are frequently neglected or annotated as variants of unknown significance. Biomolecular phase separation has increasingly appreciated roles in physiological processes and provides an additional method for interpreting disease mutations. We found that A64D and R223H on WHRN, although not affecting binding to USH2A or ADGRV1, cause deafness by disturbing WHRN dimerization and disrupting phase separation of USH2 condensates. Phase separation now turns out to be an important mechanism for interpreting hearing-related disease mutations.

## Methods
### Animal care and handling
Our research complies with all relevant ethical regulations. Mice were housed in standard laboratory cages with free access to food and water. Neonatal pups were sacrificed by rapid decapitation. The animal study was reviewed and approved by Animal Ethics Committee of Shandong University School of Life Sciences (Permission No. SYDWLL-2021-74).

### Molecular cloning
The primers were purchased from GENEWIZ, Suzhou. The DNA fragments were amplified by regular PCR (Vazyme). After purification by electrophoresis and gel extraction, the fragments were cloned into pET.32 M.3 C through homologous recombination (Yeasen, Shanghai, China). Mutants were created by PCR-based mutagenesis. DNA sequencing was provided by GENEWIZ, Suzhou.

According to conservation alignment and secondary structure prediction, different fragments were acquired. WHRN-Full Length (UniProtKB: Q9P202) was fragmented into NTD (a.a. 33-117), PDZ1 (a.a. 136-239), NPDZ1 (a.a. 33-239, including NTD and PDZ1), PDZ2 (a.a. 275-377), NPDZ12 (a.a. 33-377, includes NTD, PDZ1, and PDZ2), and PDZ3 (a.a. 710-907). PDZD7-Full Length (UniProtKB: E9Q9W7) was fragmented into PDZ12 (a.a. 84-289), PDZ1 (a.a. 81-167), PDZ2 (a.a. 202-319), and PDZ3 (a.a. 842-946). The cytoplasmic region (CT) (a.a. 5055-5193) of USH2A (UniProtKB: Q2QI47) was fragmented into USH2A-CT (a.a. 5118-5193) and USH2A-CT (Short, S; a.a. 5159-5193). The cytoplasmic region of ADGRV1 (UniProtKB: Q8VHN7) was ADGRV1-CT (a.a. 6239-6298).

### Protein expression and purification
The constructs were transformed into *Escherichia coli* BL21 (DE3) host cells at 42 °C for ~1.5 min. BL21 cells were amplified at 37 °C in a shaker until the OD values reached 0.3–0.5. The cells were then cooled to 16 °C, and IPTG was added at 250 µM to induce the protein expression. The proteins were purified using $Ni^{2+}$-NTA agarose affinity chromatography followed by size-exclusion chromatography. The buffer for size-exclusion chromatography contained 50 mM tris, PH7.5-8.0, 300 mM NaCl, 1 mM DTT, and 1 mM EDTA. The following biochemical assays were conducted in the same buffer condition.

### ITC assay
ITC assays were carried out on a MicroCal iTC200 system (Malvern Panalytical) at 25 °C. Proteins were purified as mentioned above. The purified proteins were loaded into the cell (20–50 µM) or syringe (400–800 µM). A total of 24 titrations were run. In each titration, 1 µl aliquot of proteins in the syringe was injected into the cell with a duration of 2 s. The time interval between 2 different titrations was 120 s to allow the titration peak to return to the baseline level. The data were analyzed with Origin7.0 from Microcal and fitted with a one-site binding model.

### SEC-MALS
A Superose 12 10/300 GL or Superdex 200 increase 10/300 GL column (GE Healthcare) was coupled to an AKTA FPLC system, a multi-angle static light scattering detector (miniDawn, Wyatt), and a differential refractive index detector (Optilab, Wyatt). The column was equilibrated overnight with the buffer used in protein purification. For non-complex proteins, the samples were prepared at 50 µM. For complex protein in Fig. 3b, samples were mixed with the final concentration of each component to be around 30 µM. 200–300 µl samples were filtered and loaded into the column. The elution profiles were analyzed using ASTRA 6 (Wyatt).

### FPLC coupled with fractionation
A Superose 12 10/300 GL column (GE Healthcare) was connected to the AKTA FPLC system. WHRN NPDZ12, USH2A-CT(S), and ADGRV1-CT were mixed at the molar ratio of 1:1:1. 200–300 µl samples were filtered and loaded into the column. The chromatography profiles were analyzed using Unicorn 7.0 (GE Healthcare). The fractionation samples collected were applied to SDS-PAGE and Coomassie blue staining.

### GST pull-down assay
WHRN NPDZ12 or PDZD7 PDZ12 was tagged with pET.GST. The two GST-tagged proteins were purified with GST agarose affinity chromatography followed by size-exclusion chromatography. And the potential binding partners were purified, as mentioned above. 10 µM GST proteins and 50–00 µM potential binding partners were mixed. The mixtures were incubated at room temperature for 0.5 h or at 4 °C for 1 h. Then they were centrifugated at 3000 rpm at 4 °C for 2 min. The supernatants were removed and the beads were washed with 1 ml protein purification buffer 2–3 times. Then 20 µl 2 x loading buffer was added into the tubes. Boiling at 100 °C was provided to take proteins

off the beads. Finally, the samples were applied to SDS-PAGE and Coomassie blue staining.

## Protein labeling with fluorophores

Cy3/Cy5 NHS ester (AAT Bioquest) and Alexa 405/488 NHS ester (Thermo Fisher) were dissolved using DMSO. Proteins were purified with the buffer of 50 mM tris, 300 mM NaCl, 1 mM DTT, and 1 mM EDTA, as mentioned above, and concentrated to 5-10 mg/ml. Then the buffer was changed to 100 mM NaHCO$_3$, pH 8.3, and 300 mM NaCl using a desalting column, as tris would interfere with protein labeling. The protein was again concentrated to 5-10 mg/ml and mixed with different dyes at the molar ratio of 1:1 or 1:2. The mixture was incubated in the dark at room temperature for 1 h. At the end of incubation, the buffer was changed back to 50 mM tris, 300 mM NaCl, 1 mM DTT, and 1 mM EDTA using a desalting column to quench the labeling reaction. During the second buffer change, labeling proteins were purified simultaneously. Labeling efficiencies were determined by Nanodrop 2000 (Thermo Fisher). For the labeling experiment, labeled proteins were added into the same unlabeled proteins at the molar ratio of 1%.

## Co-sedimentation-based assay

After purification, all proteins were concentrated to the required concentration. And then, they were centrifuged at 16,873 g at 4 °C for 15 min. Then the supernatants were collected into new tubes and kept on ice to avoid potential non-specific aggregates.

For the co-sedimentation-based assay, a 20 μl mixture containing 20 μM of each protein was equilibrated at room temperature for 5 min and then incubated at 30 °C for 10 min. Then the mixture was centrifuged at 16,873 g at room temperature for 10 min. The supernatant was collected and mixed with 20 μl 2 x loading buffer. The pellet was mixed with 20 μl buffer and 20 μl 2 x loading buffer to reach a volume equal to that of the supernatant. The supernatant and pellet samples were boiled at 100 °C for 10 min and applied to SDS-PAGE and Coomassie blue staining.

## Fluorescent imaging

WHRN NPDZ12, USH2A-CT, PDZD7, and ADGRV1-CT were labeled with fluorophore 488, Cy3, Cy5, and 405, respectively. Coverslips were fixed on a glass slide using double-sided tape to make flow chambers. Proteins were mixed and incubated at 30 °C for 10 min and then injected into the flow chambers. Fluorescent images were captured using Leica TCS SP8.

## FRAP assay

FRAP assay was performed on a Leica SP8 confocal microscope with LAS X software at room temperature with a 63 x oil objective. In each FRAP experiment, only one protein was labeled with Cy3 and added to the mixture at a ratio of 1%. A circular region of interest (ROI) was manually selected. And two images were taken before bleaching, with an interval time of 2 s. The bleach was achieved by 561 nm laser beam, with parameters of 100% laser power, 1.2 s dwell time and an iteration of 2 times. After bleaching, a series of images were taken in 10 min at 30 s/frame and 20 frames in total to monitor the process of fluorescence recovery. For quantitative analysis, the fluorescent intensities before and after the bleach were normalized to 100% and 0%, respectively. 3–5 of the data were collected to quantify each protein. The images were analyzed by ImageJ, and the quantification was performed by Graphpad Prism. The results were expressed as Mean ± SD. The time after bleaching and the fluorescence quantification were taken as the *X* and *Y*-axis, respectively.

## Injectoporation and immunostaining

Injectoporation was performed as previously described[21,37]. Briefly, the cochlear sensory epithelia were isolated from postnatal day 4 (P4) C57BL6/J mice of either sex (Charles River, Beijing, China; *n* = 7) and cultured in DMEM/F12 with 1.5 mg/ml ampicillin. Expression plasmids (0.5 mg/ml in Hanks' balanced salt solution) were delivered to hair cells using a glass pipette of 2 mm tip diameter. A series of three pulses at 60 V lasting 15 ms at 1-s intervals were applied to the epithelia by an electroporator (ECM Gemini X2, BTX, CA). The tissues were cultured for 18–20 h in vitro followed by sequential incubation with rabbit anti-ADGRV1 antibody (1:4000 dilution) and Alexa Fluor 594 donkey anti-rabbit secondary antibody (Thermo Fisher Scientific, Cat. No. R37119, 1:400 dilution). Stereociliary F-actin core was visualized by iFluor 405-conjugaetd phalloidin (Abcam, Cat. No. ab176752). The samples were mounted in PBS/glycerol (1:1), and images were taken using a confocal microscope with a 1.4 NA/63 Kort M27 objective lens (LSM 900, Zeiss, Germany). ADGRV1-specific antibody (1:4000 dilution) was affinity-purified from mouse and its specificity was validated using *Adgrv1* knockout mice[24].

## Alphafold protein structure prediction

The Alphafold with a deep learning algorithm was applied for complex structure prediction[38]. We fused the amino acid sequence of ADGRV1-CT (UniProtKB: Q8VHN7, a.a. 6240-6261) to the C-terminus of WHRN NPDZ1 sequence (UniProtKB: Q9P202, a.a. 33-117) for prediction. And a linker with the amino acid sequence of one repeat of Glycine-Serine is inserted between ADGRV1-CT and Whirlin NTD. The Alphafold server in Shanghai Jiaotong University (https://studio.hpc.sjtu.edu.cn) provided a high-performance computing platform. The prediction was obtained using the monomeric model of AlphaFold in default parameters. The predicted complex structure with the highest pLDDT value (87.86) was trusted with the highest confidence level and adopted for subsequent analyses[39]. And it was superimposed with the resolved complex structure of Cad23 NBM/Harmonin NTD (PDB ID: 2KBR) for validation.

## Statistics and Reproducibility

The quantification data of phase separation (Fig. 5h), and SDS-PAGE assays (Figs. 6a, 8e, Supplementary Fig. 5a) were acquired from three independent experiment. All statistics (e.g., number of biological samples (*n*) for all experiments) were described in the figure legends. One-way ANOVA was applied for experimental comparisons using GraphPad Prism. Assays were performed at least three times. All data are shown as the mean ± SD.

## Reporting summary

Further information on research design is available in the Nature Portfolio Reporting Summary linked to this article.

# Data availability

The authors declare that all data supporting the findings of this study are available within the article and its Supplementary Information files or from the corresponding author upon reasonable request. Source data are provided with this paper. The resolved complex structure of Cad23 NBM/Harmonin NTD or Harmonin NPDZ1/SANS SAM-PBM is deposited in PDB with the accession code of 2KBR or 3K1R, respectively. Source data are provided with this paper.

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

## Acknowledgements

We thank the staff members of the Large-scale Protein Preparation System and Integrated Laser Microscopy System at the National Facility for Protein Science in Shanghai (NFPS), Shanghai Advanced Research Institute, Chinese Academy of Sciences, China, for providing technical support and assistance with data collection and analysis. We thank Xiaomin Zhao and Haiyan Yu from the core facilities for life and environmental sciences, Shandong University for the technical support in confocal microscopy. We thank Dr. Jun Yang (Department of Ophthalmology and Visual Sciences, University of Utah) for providing anti-ADGRV1 antibody. This work was supported by the National Key Research and Development Program (2020YFA0509700 to Q.L., 2022YFE0131900 to Z.X.), National Natural Science Foundation of China (31900858 to Q.L., 82192861 and 82071051 to Z.X.), the "Eastern Scholar" project supported by the Shanghai Municipal Education Commission, Shandong Provincial Natural Science Foundation (ZR2020ZD39 to Z.X.), and the Interdisciplinary Program of Shanghai Jiao Tong University (project number YG2022QN064 to T.D. and Q.L.).

## Author contributions

Hu.W., H.D., R.R., T.D., L.L., Z.F., D.Z., X.W., X.Z., Ho.W., and T.D. performed experiments. Huang.W., H.D., J.S., Hao.W., Z.X., and Q.L. analyzed the data. Huang.W., H.D., Z.X., and Q.L. designed the research. Hu.W., H.D., Z.X., and Q.L. drafted the paper. All authors commented on the paper. Z.X. and Q.L. coordinated the project.

## Competing interests

The authors declare no competing interests.
