## [Peer Review File · Nature Communications]

REVIEWER COMMENTS

Reviewer #1 (Remarks to the Author):

The manuscript by Wang et al provided a detailed characterization of the interaction network of the ankle-link complex formed by several USH2 proteins and demonstrated that these components can undergo liquid-liquid phase separation to form a condensed structure. This is a significant supplement to the previous findings that the tip-link densities are shown to form using similar mechanisms. More importantly, the authors showed that some mutations found in USH2 patients, although have little impact on the interactions, severely impair the LLPS properties. This provides novel insights into the potential mechanism of USH2 pathogenesis. Overall, the study is interesting and is a strong candidate for publishing in Nature Communications. I only have a few minor comments:

1. The authors should provide the Uniprot or NCBI entries of the proteins they used for obtaining recombinant proteins in the method section.

2. The organization of the figures can be improved. For example:
Fig. 2G is about the interaction between Whirlin and Usherin/VLGR1, and should be in Fig. 1;
Fig. S2 is the pull-down assay about the interaction between PDZD7 and Usherin/VLGR1, and should be in Fig. 2.
Fig. 2E&2F can be put in the supporting materials.

3. The authors reported that the proteins were purified in a buffer containing 300 mM NaCl. Were all the assays (ITC, SEC-MALS, pull-down and LLPS) performed using the same condition? A detailed characterization of the interaction between VLGR1 and PDZD7 was recently reported (Colcombet-Cazenave et al, 2022, Front Mol Biosci), and they have different conclusions. The authors should comment on this.

4. The authors used the short version of Usherin CT to test its interaction with PDZD7. What about the WT Usherin CT? Can it bind to PDZD7 in the same manner (stoichiometry and affinity)?

5. The corresponding MALS fitting results for Usherin-CT(S)/Whirlin/VLGR1-CT (Fig. 3B-b4, bottom) should be shown.

6. It seems that 'Whirlin' in the pull-down, SEC-MALS and LLPS assays was Whirlin NPDZ12 but not full-length Whirlin. The authors should change the name to avoid misunderstanding.

7. For the sedimentation assays, it is better to quantify by densitometry, especially for those key experiments (Whirlin/Usherin at different concentrations, disruption of LLPS by adding different concentrations of VLGR1).

8. For the FRAP assay (Fig. 5H), the authors should add the original movie or the images at different time points.

9. Full-length Whirlin alone was shown to undergo LLPS. How about full-length Whirlin in this ankle-link complex system? The authors should also comment on it or provide experimental evidence.

10. Where is T110 in the structure? The authors should also show it in Fig. 8A.

11. The authors should add more details about how the prediction of the Whirlin/VLGR1 complex structure was performed. Did the authors use Alphafold multimer mode? What sequences were used as input? Such information should be

included in the method section.

12. Also, the reliability of the predicted structure should better be validated, preferably by mutagenesis-based binding assays.

Reviewer #2 (Remarks to the Author):

The manuscript from Wang et al. provides a plausible explanation for how ankle link complexes form in hair cells. Comprehensive ITC and SEC-MALS experiments allowed for the identification of key domains of each of the proteins that bind to each other; the interaction mapping carried out here is very believable and is rigorously done.

Major issues

1. The manuscript badly needs editing from an English-speaking reader. While the erratic grammar does not prevent one from fully understanding the text, it makes it quite a struggle. There are funny turns of phrase, missing articles, plural problems, etc. I could spend hours and hours correcting the grammar but I am unwilling to.

2. The hair-cell localization experiments of Fig. 7 are inadequate. With top-down views of hair bundles at very low resolution, it is impossible to determine where the fluorescent proteins are located. These experiments are critical for the paper—they tie the in vitro biochemistry to hair cell biology—but far better microscopy and more comprehensive experiments are required.

3. The statement that “[e]lectron densities have been observed at...[the] ankle link region” is not supported by the references used and in fact does not seem to be true. Reference 13 does not include any transmission electron microscopy, required for visualizing the electron dense structures of hair bundles. Reference 22 has TEM of photoreceptors but only SEM of hair cells. Goodyear et al. (2005), a good reference for ankle links, shows some increased electron density in the region of ankle links but that looks extracellular, i.e., because of the density of ADGRV1 and USH2A. I am not aware of any papers clearly showing electron density underneath the ankle links—unlike what is seen with the upper and lower tip link densities.

4. Is this really liquid-liquid phase separation? The slow (ADGRV1 and USH2A) or absent (WHRN and PDZD7) FRAP recovery suggests that a solid phase is formed, not liquid. It is important what it is called, as separations into liquid phase or solid phase lead to different biophysical properties. Solid phase is OK, however, for a structural/scaffolding function for the phenomenon you are investigating.

Other issues

5. Strongly suggest that you use all official protein symbols, especially WHRN instead of Whirlin, USH2A instead of Usherin, and ADGRV1 instead of VLGR1. Consistency in symbols is very important, and I recommend using the protein versions (all caps, no italics) of the official gene symbols. I realize that the names you use are common, but there are many names for the same protein and it's important to use systematic databases.

6. Please be thoughtful in reporting significant digits. For example, “66.23 ± 35.17 μM” makes no sense at all, as the number of significant digits indicate precision far, far beyond what the error measurement suggests is a limit. In this example, I would report “66 ± 35 μM” (or even “70 ± 40 μM”). The error value should not have more than two significant digits (not two decimal places), and that should

drive reporting of the mean value.

7. Page 3, line 98. This statement needs a reference.

Reviewer #3 (Remarks to the Author):

This study provides interesting molecular-level insights into the assembly of the ankle link complex of stereocilia. Through a series of binding and other experiments, they localize relevant interaction domains and characterize the effects of certain residue substitutions. While the results are of interest to the community, more detail regarding methods and results are needed, as noted in the specific comments below.

1. Line 126: "Whirlin NTD bears a high sequence similarity of 60.98%...". Sequence similarities do not have numerical values percentage values; the authors likely mean "sequence identity" and should change the wording in this sentence accordingly. Otherwise, the numerical value should be removed.

2. In Figure 6E, the authors label the right-most panel with the "VLGR1-CT NBM/Whirlin NTD". Although the figure legend notes that this is a model, to make this more clear to readers that are viewing the figure, the authors should clearly note that it is a model in the panel title, e.g.: (AlphaFold) "Model of VLGR1-CT...".

3. The description of the AlphaFold modeling procedure (lines 563-570) requires more information. A fusion is described in this section, but it notes Usherin-CT rather than VLGR1-CT, which does not seem to be in accordance with the reported model. The authors should clearly note the specific residue ranges and proteins used as input to AlphaFold, and for the fusion they should note whether a linker was used (which would be expected, to permit degrees of freedom for AlphaFold modeling), and if a linker was used, the authors should describe its length and constitution (e.g. poly-Glycine). Additionally, they should note whether default AlphaFold parameters were used, and whether AlphaFold-Multimer or AlphaFold (monomeric model) were used to model the complex.

4. The authors note that pLDDT confidence scores were used to select the AlphaFold model, but they do not seem to report the actual pLDDT score of the model that they report. Both pLDDT and pTM AlphaFold model confidence scores should be noted by the authors in the Results and/or Figure 6 legend, to provide readers with a better understanding of the model's confidence.

5. The representation of the Whirlin-NTD cartoon in Figure 8A (right-hand part) seems clipped in the superposition with the Harmonin-NTD, resulting in multiple apparent breaks in the helix. The authors should try to adjust this representation, as the helix breaks are distracting and potentially misleading.

REVIEWER COMMENTS

Reviewer #1 (Remarks to the Author):

The manuscript by Wang et al provided a detailed characterization of the interaction network of the ankle-link complex formed by several USH2 proteins and demonstrated that these components can undergo liquid-liquid phase separation to form a condensed structure. This is a significant supplement to the previous findings that the tip-link densities are shown to form using similar mechanisms. More importantly, the authors showed that some mutations found in USH2 patients, although have little impact on the interactions, severely impair the LLPS properties. This provides novel insights into the potential mechanism of USH2 pathogenesis. Overall, the study is interesting and is a strong candidate for publishing in Nature Communications. I only have a few minor comments:

1. The authors should provide the Uniprot or NCBI entries of the proteins they used for obtaining recombinant proteins in the method section.

Thank you for reminding us of this point. The Uniprot entries of Whirlin, PDZD7, Usherin, and VLGR1 are Q9P202, E9Q9W7, Q2QI47, and Q8VHN7, respectively. We have added the entry information in the section on molecular cloning.

2. The organization of the figures can be improved. For example:

Fig. 2G is about the interaction between Whirlin and Usherin/VLGR1, and should be in Fig. 1;

Fig. S2 is the pull-down assay about the interaction between PDZD7 and Usherin/VLGR1, and should be in Fig. 2.

Fig. 2E&2F can be put in the supporting materials.

Thank you for providing the suggestions on Figure organization. We have made the revisions according to your suggestion to make it easy to read.

3. The authors reported that the proteins were purified in a buffer containing 300 mM NaCl. Were all the assays (ITC, SEC-MALS, pull-down and LLPS) performed using the same condition?

All the assays were performed using the buffer containing 300 mM NaCl. To make it clear, detailed information was added in the section of "Protein expression and purification".

A detailed characterization of the interaction between VLGR1 and PDZD7 was recently reported (Colcombet-Cazenave et al, 2022, Front Mol Biosci), and they have different conclusions. The authors should comment on this.

Thank you for reminding us. The interaction between VLGR1 and PDZD7 was characterized using fluorescence titration in the above-mentioned paper. We have cited the paper and described our opinion in Result (Page 4, Line 156-159).

4. The authors used the short version of Usherin CT to test its interaction with PDZD7.

What about the WT Usherin CT? Can it bind to PDZD7 in the same manner (stoichiometry and affinity)?

Thank you for raising this question. We performed the ITC assay to measure the binding affinity between the WT Usherin CT and PDZD7. As expected, the WT Usherin-CT binds to PDZD7 in a similar affinity as Usherin-CT(S) ($K_d=11$ vs $15 \mu\text{M}$) (panel A below). We also used SEC-MALS to elucidate the stoichiometry of the PDZD7/Usherin-CT interaction. Usherin-CT appears as a dimer in solution. And when it is mixed with Trx-tagged PDZD7 PDZ1, the molecular size of the complex is calculated to be 77.5 kDa, which is fitted as a stoichiometry of 2:2 (panel B below).

Characterization of the binding between USH2A and PDZD7. (A) Binding affinities of USH2A-CT (left panel) or USH2A-CT(S) (right panel) to PDZD7 PDZ12 determined by ITC. (B) SEC-MALS shows the molecular size of USH2A-CT in mixture with trx-PDZD7 PDZ1.

5. The corresponding MALS fitting results for Usherin-CT(S)/Whirlin/VLGR1-CT (Fig. 3B-b4, bottom) should be shown.

We thank the reviewer for noting this issue for us. We have added the MALS data of Usherin-CT(S)/Whirlin/VLGR1-CT in Fig. 3B, b4.

Fig. 3B (b4). SEC-MALS shows the molecular size of USH2A-CT(S)/WHRN/ADGRV1-CT. WHRN NPDZ12 is abbreviated as WHRN.

6. It seems that ‘Whirlin’ in the pull-down, SEC-MALS and LLPS assays was Whirlin NPDZ12 but not full-length Whirlin. The authors should change the name to avoid misunderstanding.

We are sorry for the ambiguity. For Figure 3 and the following Figures, the fragments of Whirlin NPDZ12 and PDZD7 PDZ12 were mostly used. To make it more clear and avoid misunderstanding, we have changed the names to make it easier to read.

7. For the sedimentation assays, it is better to quantify by densitometry, especially for those key experiments (Whirlin/Usherin at different concentrations, disruption of LLPS by adding different concentrations of VLGR1).

We have made the quantifications of the co-sedimentation assays following the reviewers' comments. We have added a sub-panel in Figure S5A (bottom panel), showing the concentration-dependent protein condensation of WHRN/Usherin-CT at 10-20 μM (presented below).

Fig. 5SA, Bottom panel. Quantifications of the percentage of WHRN or USH2A-CT recovered from pellets in the co-sedimentation described in up panel.

Also, Figure S7A is added to quantify the disruption of WHRN/USH2A-CT (20+20 μM) LLPS by adding different concentrations of ADGRV1-CT (0-60 μM) (see below).

Fig. S7A. Quantifications of the percentage of WHRN or USH2A-CT recovered from pellets in the co-sedimentation assays described in Fig. 6A.

8. For the FRAP assay (Fig. 5H), the authors should add the original movie or the images at different time points.

Thank you for the suggestion. We have now added the original images in Fig. S6C.

Fig. S6C. The original movie of the FRAP analysis described in Fig. 5H. Scale bar: 1 μ m. WHRN NPDZ12 and PDZD7 PDZ12 is abbreviated as WHRN and PDZD7, respectively.

9. Full-length Whirlin alone was shown to undergo LLPS. How about full-length Whirlin in this ankle-link complex system? The authors should also comment on it or provide experimental evidence.

Thank you for raising this important question for us. Full-length Whirlin alone was reported to undergo LLPS in Lin, L et. al 2020. Therefore, we speculate that the threshold concentration for LLPS of the Whirlin/USH2 protein complex would be lowered if the full-length Whirlin protein is used. We have made the comments in the section of “LLPS in hearing system” in Discussion (Page 9, line 391-395).

10. Where is T110 in the structure? The authors should also show it in Fig. 8A.

Thank you for reminding us. We have marked T110 in WHRN NPDZ1 and found it to localize on the α E helix of WHRN NTD.

Fig. 8A. The predicted WHRN NPDZ1 structure is overlaid onto the atomic structure of Harmonin NPDZ1/SANS SAM-PBM (PDB accession code 3K1R). Superimposition shows that A64, T110, and R223 on WHRN NPDZ1 correspond to M32, T78, and L169 on Harmonin NPDZ1, respectively. The side chain of WHRN A64 or Harmonin M32 point outwards, making it possible to mediate NTD oligomerization by hydrophobic interaction. Thus, A64D may disturb WHRN dimerization by disrupting WHRN NTD oligomerization. WHRN T110 or Harmonin T78 localizes to the α E helix of NTD. We speculate that the T110A may affect WHRN's interaction with USH2A by disrupting the overall conformation of WHRN. WHRN R223 or Harmonin L169 localizes on the mini-domain. The mini-domain on Harmonin helps the folding of NPDZ1 supramodule. We speculate that R223H disrupts the overall folding of the mini-domain and further leads to the misfolding of NPDZ1 on WHRN.

11. The authors should add more details about how the prediction of the Whirlin/VLGR1 complex structure was performed. Did the authors use AlphaFold multimer mode? What sequences were used as input? Such information should be included in the method section.

Thank you for providing the essential suggestions on the structure prediction. We have added methodology details on structure prediction in the section of “AlphaFold protein structure prediction” in Methods. The amino acid sequence of VLGR1-CT (UniProtKB: Q8VHN7, a.a. 6240-6261) is fused to the C-terminus of Whirlin NTD sequence (UniProtKB: Q9P202, a.a. 33-117) for prediction. And a linker with the amino acid sequence of Glycine-Serine is inserted between VLGR1-CT and Whirlin NTD to provide freedom for AlphaFold modeling. The prediction was obtained using the monomeric model of AlphaFold in default parameters.

12. Also, the reliability of the predicted structure should better be validated, preferably by mutagenesis-based binding assays.

Thanks for the suggestion. To validate the reliability of the predicted structure of ADGRV1-CT NBM/WHRN NTD, we truncated NBM from ADGRV1-CT for ITC assays. Expectedly, truncating NBM from ADGRV1-CT abolished its binding to WHRN NTD. And compared with the WT protein, ADGRV1-CT Δ NBM showed decreased binding affinity to WHRN NPDZ12, as the PBM contributed partly to the interaction. We have added the mutagenesis-based ITC assays in Figure S7C (presented

below).

Fig. S7C. Up panel, ITC assays showing that truncating NBM (Δ NBM) from ADGRV1-CT abolished and diminished its bindings to WHRN NTD and WHRN NPDZ12, respectively; bottom panel, summaries of the binding affinities between WT or Δ NBM constructs of ADGRV1-CT and NTD or NPDZ12 of WHRN. NBM, NTD binding motif. NTD, N-terminal domain.

Reviewer #2 (Remarks to the Author):

The manuscript from Wang et al. provides a plausible explanation for how ankle link complexes form in hair cells. Comprehensive ITC and SEC-MALS experiments allowed for the identification of key domains of each of the proteins that bind to each other; the interaction mapping carried out here is very believable and is rigorously done.

Major issues

1. The manuscript badly needs editing from an English-speaking reader. While the erratic grammar does not prevent one from fully understanding the text, it makes it quite a struggle. There are funny turns of phrase, missing articles, plural problems, etc. I could spend hours and hours correcting the grammar but I am unwilling to.

Thank you for the suggestion. We have taken extra care in correcting the errors in grammar and polishing the language in the revised manuscript. The American Journal Expert (AJE) helped us with the language editing.

2. The hair-cell localization experiments of Fig. 7 are inadequate. With top-down views of hair bundles at very low resolution, it is impossible to determine where the fluorescent proteins are located. These experiments are critical for the paper—they tie the in vitro biochemistry to hair cell biology—but far better microscopy and more comprehensive experiments are required.

We thank the reviewer for noting this issue for us. We consolidated our hair cell localization experiments and updated the images in Figure 7 (presented below for the convenience of viewing). First, to get high-resolution images, we re-injectoprated the expression plasmids, encoding full-length WHRN or WHRN without NTD or PDZ1 domain, into hair cells and examined their localization using a confocal microscope. Then, in order to link the biochemical and structural characterization, human genetic data, and hair-cell experiments together, we also examined the localization of the deafness mutations (EGFP-WHRN-R223H and EGFP-WHRN-A64D) in hair cells. The results reveal that the two-point mutations show a weak effect on the localization of WHRN in the stereocilia (Fig. 7D and E), consistent with our biochemical data and structural analysis.

Fig. 7

Fig. 7. NTD and PDZ1 are necessary for WHRN to target to the ankle region of stereocilia. Cochlear explants from P4 wild-type mice were injectoprated with expression vectors to express EGFP-tagged full-length WHRN (A), WHRN lacking NTD (B), WHRN lacking PDZ1 (C), WHRN-R223H (D), or WHRN-A64D (E) in cochlear hair cells. Immunostaining with anti-ADGRV1 antibody was performed to locate ADGRV1 (red). Phalloidin staining was performed to indicate the stereocilia (blue). Scale bars: 2 μ m. NTD, N-terminal domain.

3. The statement that “[e]lectron densities have been observed at...[the] ankle link region” is not supported by the references used and in fact does not seem to be true. Reference 13 does not include any transmission electron microscopy, required for

visualizing the electron dense structures of hair bundles. Reference 22 has TEM of photoreceptors but only SEM of hair cells. Goodyear et al. (2005), a good reference for ankle links, shows some increased electron density in the region of ankle links but that looks extracellular, i.e., because of the density of ADGRV1 and USH2A. I am not aware of any papers clearly showing electron density underneath the ankle links—unlike what is seen with the upper and lower tip link densities.

We thank the reviewer for noting this issue for us. We fully agree with the reviewer that TEM Images reported by Goodyear et al. (2005) show some increased electron density in the extracellular region of ankle links. But we also noticed the darker color along and beneath the plasma membrane in the ankle-link region (presented below, panel A). A similar dark density is also reported by McGee, J et al. (2006), providing reliable evidence for our search on the phase separation of the USH2 protein complex (presented below, panel B). We also used electron microscopy to capture the ankle-links of OHC in P2 and observed similar weak enrichment at the ankle-link region, including the extracellular, along, and beneath the plasma membrane.

Goodyear et al. (2005) (A) and McGee, J et al. (2006) (B). Scale bars=200 nm.

We have now sorted out our references and rephrased this point in our revised version as “Notably, compared to the upper and lower tip-link densities, a weaker electron-dense structure can be also observed at the ankle-link region of stereocilia”. This kind of weak density formation is consistent with the temporal assembly property of ankle-link for hair bundle development. The darker densities of the upper and lower tip-link indicate their structural function as rigid anchors, which is similar to the post and pre-synapse density in neurons. It would be interesting to assess the physical properties of different kinds of condensate. For ankle-link condensate, reconstitution of transmembrane proteins that cluster on the supported lipid bilayer can be applied to study these microclusters.

4. Is this really liquid-liquid phase separation? The slow (ADGRV1 and USH2A) or absent (WHRN and PDZD7) FRAP recovery suggests that a solid phase is formed, not liquid. It is important what it is called, as separations into liquid phase or solid phase lead to different biophysical properties. Solid phase is OK, however, for a structural/scaffolding function for the phenomenon you are investigating.

We thank the reviewer for noting this issue. We fully agree with the reviewer that it is important to clarify whether the condensate is more liquid-like or solid-like. So, we combined fluorescence imaging and FRAP assay to describe the phenomenon of phase separation.

First, we monitored the phase droplets under the microscope. The spherical morphology of condensate droplets and their fusion event upon contact suggests the condensates are liquid-like. We have added a sub-panel in Figure S6 (panel B), showing the small droplets gradually fuse into larger ones, reinforcing the conclusion of LLPS. For convenience, we have included the panel below.

Fig. S6B. The small droplets fused into larger ones. Scale bar: 1 μm . WHRN NPDZ12 and PDZD7 PDZ12 are abbreviated as WHRN and PDZD7, respectively.

FRAP analysis is increasingly adopted to demonstrate the mobility and dynamics of molecules within liquid droplets. Molecule exchange within the condensed phase or between the condensed and diluted phases can be captured by FRAP experiments depending on different degrees of bleaching (half bleach or whole bleach). The slow recovery in this study may result from the high photobleaching intensity of the laser, which may engender heat and prevent protein molecules in the surrounding solution from entering the droplets. So, we repeated the FRAP analysis but with different laser intensities for photobleach (100% laser power x 20 times, 100% laser power x 2 times, and 10% laser power x 2 times). The FRAP assay was performed on a Leica SP8 confocal microscope at room temperature with a 63 x oil objective. In each FRAP experiment, only one protein was labeled with Cy3 and added to the mixture at a ratio of 1%. A circular region of interest (ROI) with diameters of 2-3 μm was manually selected for bleach by a 561 nm laser beam. The images were taken in 10 min at 30 s/frame and 30 frames in total. Under different laser intensities, we find that the recovery after photobleaching of both proteins correlates negatively with the intensity of photobleaching.

FRAP assay (A) and PDZD7 (B) under different laser intensities for photobleach (100% laser power x 20 times, 100% laser power x 2 times, and 10% laser power x 2 times). The results

were presented as means \pm SDs with 3-5 droplets analyzed for each protein. WHRN NPDZ12 and PDZD7 PDZ12 were abbreviated as WHRN and PDZD7, respectively.

We then photobleached USH2A-CT and ADGRV1-CT at the intensity of 10% x 2 times and obtained a new FRAP result with a much higher recovery rate. It is shown that all four proteins appear to recover more quickly than they were photobleached with 100% intensity x20 times (as performed in the original assays). Thus, the newly-conducted FRAP assays demonstrated that the USH2 protein complex in this study really undergoes liquid-liquid phase separation. The new Figure for the FRAP assay is added in Figure 5H.

Fig. 5H. Kinetic quantification of the FRAP assay showing the recovery rates of WHRN, USH2A-CT, ADGRV1-CT, and PDZD7. The results were presented as means \pm SDs with 6 droplets analyzed for each protein. WHRN NPDZ12 and PDZD7 PDZ12 were abbreviated as WHRN and PDZD7, respectively.

We also summarized some reported recovery rates below. However, due to a lack of detailed descriptions of the laser parameters for photo bleach, we are not able to correlate the recovery rates with the photobleaching intensity, but still, the table shows a broad range of recovery rates from 15% to 65%.

Protein	Recovery	Function	Reference PMID
bFGF	55%	Signaling transduction	35236856
Whirlin, Gpsm2	15%, 20%	Actin bundling	33626355
TPX2	25%	Spindle formation	31937751
Par3	40%	Cell polarity establishment	32385244
USH1C, ANKS4B, M7BNC	40%, 60%, 65%	Stabilizing tip-links in microvilli	31644917

Other issues

5. Strongly suggest that you use all official protein symbols, especially WHRN instead of Whirlin, USH2A instead of Usherin, and ADGRV1 instead of VLGR1. Consistency in symbols is very important, and I recommend using the protein versions (all caps, no italics) of the official gene symbols. I realize that the names you use are common, but there are many names for the same protein and it's important to use systematic

databases.

Thank you for the suggestion. Whirlin, Usherin, and VLGR1 have been revised as WHRN, USH2A, and ADGRV1, respectively.

6. Please be thoughtful in reporting significant digits. For example, “ $66.23 \pm 35.17 \mu\text{M}$ ” makes no sense at all, as the number of significant digits indicate precision far, far beyond what the error measurement suggests is a limit. In this example, I would report “ $66 \pm 35 \mu\text{M}$ ” (or even “ $70 \pm 40 \mu\text{M}$ ”). The error value should not have more than two significant digits (not two decimal places), and that should drive reporting of the mean value.

Thank you for your kind suggestion. We are sorry for the sloppiness. According to your suggestion, we rephrase the mean value for all the K_d values to take two significant digits.

7. Page 3, line 98. This statement needs a reference.

We are sorry for missing the references. There are two papers reporting the LLPS-mediated formation of USH1 condensates. In 2019, He, Y. et. al discovered the formation of MYO7A/USH1C/USH1G condensates through LLPS. And in 2021, Lin, L et. al reported the formation of Whirlin-Myo15-Eps8 condensates through LLPS. Both two papers are cited following the statement in the manuscript now.

Reviewer #3 (Remarks to the Author):

This study provides interesting molecular-level insights into the assembly of the ankle link complex of stereocilia. Through a series of binding and other experiments, they localize relevant interaction domains and characterize the effects of certain residue substitutions. While the results are of interest to the community, more detail regarding methods and results are needed, as noted in the specific comments below.

1. Line 126: “Whirlin NTD bears a high sequence similarity of 60.98%...”. Sequence similarities do not have numerical values percentage values; the authors likely mean “sequence identity” and should change the wording in this sentence accordingly. Otherwise, the numerical value should be removed.

We thank the reviewer for his/her advice and we have now modified the text: “The amino acid sequence of WHRN NTD is highly similar to Harmonin NTD...” (Page 3, Line 115-116)

2. In Figure 6E, the authors label the right-most panel with the “VLGR1-CT NBM/Whirlin NTD”. Although the figure legend notes that this is a model, to make this more clear to readers that are viewing the figure, the authors should clearly note that it is a model in the panel title, e.g.: (AlphaFold) “Model of VLGR1-CT...”.

We have modified the panel title as “Model of ADGRV1-CT NBM/WHRN NTD” in the revised paper.

3. The description of the AlphaFold modeling procedure (lines 563-570) requires more information. A fusion is described in this section, but it notes Usherin-CT rather than VLGR1-CT, which does not seem to be in accordance with the reported model. The authors should clearly note the specific residue ranges and proteins used as input to AlphaFold, and for the fusion they should note whether a linker was used (which would be expected, to permit degrees of freedom for AlphaFold modeling), and if a linker was used, the authors should describe its length and constitution (e.g. poly-Glycine). Additionally, they should note whether default AlphaFold parameters were used, and whether AlphaFold-Multimer or AlphaFold (monomeric model) were used to model the complex.

Thank you for the suggestions above. VLGR1-CT is used for sequence fusion in the model prediction, we are extremely sorry for the typo mistake in the draft. And we have added the missing methodology information in the section of “AlphaFold protein structure prediction” in Methods. The amino acid sequence of VLGR1-CT (UniProtKB: Q8VHN7, a.a. 6240-6261) is fused to the C-terminus of Whirlin NTD sequence (UniProtKB: Q9P202, a.a. 33-117) for prediction. And a linker with the amino acid sequence of Glycine-Serine is inserted between VLGR1-CT and Whirlin NTD to provide freedom for AlphaFold modeling. The prediction was obtained using the monomeric model of AlphaFold in default parameters.

4. The authors note that pLDDT confidence scores were used to select the AlphaFold

model, but they do not seem to report the actual pLDDT score of the model that they report. Both pLDDT and pTM AlphaFold model confidence scores should be noted by the authors in the Results and/or Figure 6 legend, to provide readers with a better understanding of the model's confidence.

Sorry for the missing information. In the revised manuscript, we added the pLDDT score in both Methods and Result sections. The predicted structure was generated by the monomer mode with default parameters which only provide pLDDT but not pTM value. The pLDDT score is 87.86 for ADGRV1-CT NBM/WHRN NTD.

We also conducted a biochemistry assay to validate the reliability of the predicted structure of ADGRV1-CT NBM/WHRN NTD. We truncated NBM from ADGRV1-CT for ITC assays. Truncating NBM from ADGRV1-CT abolished its binding to WHRN NTD. And compared with the WT protein, ADGRV1-CT Δ NBM showed decreased binding affinity to WHRN NPDZ12, as the PBM contributed partly to the interaction. We have added the mutagenesis-based ITC assays in Figure S7C (presented below).

Fig. S7C. Up panel, ITC assays showing that truncating NBM (Δ NBM) from ADGRV1-CT abolished and diminished its bindings to WHRN NTD and WHRN NPDZ12, respectively; bottom panel, summaries of the binding affinities between WT or Δ NBM constructs of ADGRV1-CT and NTD or NPDZ12 of WHRN. NBM, NTD binding motif. NTD, N-terminal domain.

5. The representation of the Whirlin-NTD cartoon in Figure 8A (right-hand part) seems clipped in the superposition with the Harmonin-NTD, resulting in multiple apparent breaks in the helix. The authors should try to adjust this representation, as the helix breaks are distracting and potentially misleading.

Thank you for reminding us. We have adjusted the Figure to avoid breaks.

Fig. 8A. The predicted WHRN NPDZ1 structure is superimposed with the resolved structure of Harmonin NPDZ1/SANS SAM-PBM (PDB accession code 3K1R).

REVIEWERS' COMMENTS

Reviewer #1 (Remarks to the Author):

The authors have addressed all of my concerns.

Reviewer #2 (Remarks to the Author):

Reviewer #2, review of rebuttal

1. Fixed.

2. Based on Figure 7, while I agree that delta-NTD localization at ankles is reduced, I do not agree with the authors' conclusion that delta-PDZ1, R223J, and A640 mutants have altered ankle link localization. These are very important experiments, and I know that they are difficult—the authors should be commended for trying. However, injectoporation leads to considerable variations in expression levels of plasmids, and the tip labeling to ankle labeling variations the authors see could be due to stoichiometry. For these experiments to be convincing, assuming that a subtle change in distribution is what the result is, then the authors would need to quantify the relative amounts of labeling in the soma, tips, and ankles—not easy to do. I would therefore not recommend including panels C-E, and instead include more examples of full length and delta-NTD.

3. I'm not convinced by the argument, but the authors did make me look more closely. I am OK with leaving the statement as long as it is referenced by the two papers discussed.

4. The new FRAP data are a big improvement and strongly make the case for liquid-liquid phase separation.

5. Fixed.

6. Fixed.

7. Fixed.

Reviewer #3 (Remarks to the Author):

The authors have addressed most of the comments satisfactorily. However, the description of the linker used between the proteins for modeling is too vague as written (line 555): "And a linker with the amino acid sequence of Glycine-Serine is inserted...". The length of the linker should be given (e.g. 20 amino acids, 10 repeats of Gly-Ser, etc.) so that readers will

understand the length of the linker used for the modeling. As currently written, it appears that only one Glycine-Serine (two residues) was used for the linker.

Reviewer #1 (Remarks to the Author):

The authors have addressed all of my concerns.

Thanks.

Reviewer #2 (Remarks to the Author):

Reviewer #2, review of rebuttal

1. Fixed.

Thanks.

2. Based on Figure 7, while I agree that delta-NTD localization at ankles is reduced, I do not agree with the authors' conclusion that delta-PDZ1, R223J, and A640 mutants have altered ankle link localization. These are very important experiments, and I know that they are difficult—the authors should be commended for trying. However, injectoporation leads to considerable variations in expression levels of plasmids, and the tip labeling to ankle labeling variations the authors see could be due to stoichiometry. For these experiments to be convincing, assuming that a subtle change in distribution is what the result is, then the authors would need to quantify the relative amounts of labeling in the soma, tips, and ankles—not easy to do. I would therefore not recommend including panels C-E, and instead include more examples of full length and delta-NTD. Thanks. Under your advice, we have put more figures describing the localizations of full-length and delta-NTD WHRN, while results concerning delta-PDZ1, R223J, and A640 mutants have been removed.

Fig. 7. NTD is necessary for WHRN to target the ankle region of stereocilia. Cochlear explants from P4 wild-type mice were injectoparated with expression vectors to express EGFP-tagged full-length WHRN (a) or WHRN lacking NTD (b) in cochlear hair cells. Immunostaining with an anti-ADGRV1 antibody was performed to locate ADGRV1 (red). Phalloidin staining was performed to indicate the stereocilia (blue). Scale bars: 2 μ m. NTD, N-terminal domain.

3. I'm not convinced by the argument, but the authors did make me look more closely. I am OK with leaving the statement as long as it is referenced by the two papers discussed.

Thanks.

4. The new FRAP data are a big improvement and strongly make the case for liquid-liquid phase separation.

Thanks.

5. Fixed.

Thanks.

6. Fixed.

Thanks.

7. Fixed.

Thanks.

Reviewer #3 (Remarks to the Author):

The authors have addressed most of the comments satisfactorily. However, the description of the linker used between the proteins for modeling is too vague as written (line 555): "And a linker with the amino acid sequence of Glycine-Serine is inserted...". The length of the linker should be given (e.g. 20 amino acids, 10 repeats of Gly-Ser, etc.) so that readers will understand the length of the linker used for the modeling. As currently written, it appears that only one Glycine-Serine (two residues) was used for the linker.

It has now been noted that one repeat of Glycine-Serine is used for the linking.